# Practical Near Neighbor Search via Group Testing

**Joshua Engels**[*]
Department of Computer Science
Rice University
Houston, Texas, USA
jae4@rice.edu

**Benjamin Coleman**[*]
Electrical and Computer Engineering
Rice University
Houston, Texas, USA
ben.coleman@rice.edu

**Anshumali Shrivastava**
Department of Computer Science
Rice University & ThirdAI Corp.
Houston, Texas, USA
anshumali@rice.edu

## Abstract

We present a new algorithm for the approximate near neighbor problem that combines classical ideas from group testing with locality-sensitive hashing (LSH). We reduce the near neighbor search problem to a group testing problem by designating neighbors as "positives," non-neighbors as "negatives," and approximate membership queries as group tests. We instantiate this framework using distance-sensitive Bloom Filters to Identify Near-Neighbor Groups (FLINNG). We prove that FLINNG has sub-linear query time and show that our algorithm comes with a variety of practical advantages. For example, FLINNG can be constructed in a single pass through the data, consists entirely of efficient integer operations, and does not require any distance computations. We conduct large-scale experiments on high-dimensional search tasks such as genome search, URL similarity search, and embedding search over the massive YFCC100M dataset. In our comparison with leading algorithms such as HNSW and FAISS, we find that FLINNG can provide up to a 10x query speedup with substantially smaller indexing time and memory.

## 1 Introduction

Nearest neighbor search is a fundamental problem with many applications in machine learning systems. Informally, the task is as follows. Given a dataset $D = \{x_1, x_2, ...x_N\}$, we wish to build a data structure that can be queried with any point $q$ to obtain a small set of points $x_i \in D$ that have high similarity (low distance) to the query. This structure is called an *index*. Near neighbor indices form the backbone of production models in recommendation systems, social networks, genomics, computer vision and many other application domains.

**Applications:** In this paper, we focus on algorithms for approximate near neighbor search over high-dimensional large scale datasets. Such tasks frequently arise in genomics, web-scale data mining, machine learning, and other large-scale applications. Consider the Yahoo Flickr Creative Commons dataset (YFCC100M) which consists of 100 million media embeddings that are derived from the neuron activations for a convolutional neural network [21]. Each embedding is a 4096-dimensional vector. The dataset is about 1TB in size and presents a substantial challenge for even the most popular algorithms, which struggle with memory, index construction, and query time. Similar issues occur

---

[*]Equal contribution.

35th Conference on Neural Information Processing Systems (NeurIPS 2021).

in genomics, where the task is to identify genome sequences with a high Jaccard similarity to the query. Modern genomic datasets can contain millions of reads with billions of possible $n$-gram sequences [17]. Many algorithms work well when there are a few hundred dimensions but are inappropriate for such applications. Our experiments demonstrate that for the datasets of interest in this paper, popular indices like HNSW and FAISS can take days to build, require gigabytes of RAM and have a suboptimal precision-recall-query time tradeoff.

Since our goal is to perform approximate search, dimensionality reduction is a reasonable strategy. However, dimensionality reduction is costly for ultra-high dimensional data. In genomics applications, $n$-gram sizes are typically very large ($n > 18$). Thus, the one-hot encoding of each sequence can require billions of dimensions ($4^{18} \approx 68B$), making it intractable to learn an embedding model. For embedding applications such as YFCC100M or product search, a large embedding dimension can lead to performance improvements [15]. Dimensionality reduction can incur a performance penalty, so we may wish to perform the near neighbor search over the original metric space.

Ideally, we would choose an algorithm that did not store data points in RAM, evaluate the distance function many times, employ iterative and non-streaming processes such as $k$-means, or construct complicated structures such as graphs, which are hard to parallelize and distribute. Recent algorithms such as FLASH [22] provide the ability to search based on aggregate LSH count statistics without computing distances, but these methods are heuristics that do not have theoretical guarantees. On the other hand, algorithms such as LSH, which have a well-established theoretical grounding, tend to perform poorly in practice because of their prohibitive hash table size and post filtering stage, which needs many ($N^\rho$ in theory) distance computations. In this paper, we present an algorithm having all the practical advantages of a system like FLASH while also being more accurate, theoretically sound, and provably sub-linear.

## 1.1  Our Contribution

In this paper, we address the computational challenges of high-dimensional similarity search by presenting an index with fast construction time, low memory requirement, and zero query-time distance computations. Our approach is to transform a near neighbor search problem into a group testing problem by designing a test that outputs "positive" when a group of points contains a near neighbor to a query. That is, each test answers an approximate membership query over its group. Given a query, our algorithm produces a $B \times R$ array of group test results that can be efficiently decoded to identify the nearest neighbors. This is more efficient than statistical aggregation algorithms like FLASH because each test filters out entire groups of non-neighbors with a single test operation.

We develop a concrete example of such an algorithm by using Filters to Identify Near Neighbor Groups (FLINNG). We use a standard non-adaptive group testing design with distance-sensitive Bloom filters as tests. We prove that FLINNG solves the randomized nearest neighbor problem in $O\left(\log^2(\frac{1}{\delta})\log^3(N)N^{\frac{1}{2}+\gamma}\right)$ time, where $\gamma$ is a query-dependent parameter that characterizes query stability. We also implement FLINNG in C++ and conduct experiments on real-world high-dimensional datasets from genomics, embedding search, and URL analysis, where FLINNG achieves up to a 10x query speedup over existing indices with faster construction time and lower memory.

## 2  Related Work

The near neighbor problem has been the focus of more than four decades of intense research activity. The low dimensional problem is particularly well-understood, with space partitioning trees that can efficiently find the exact $k$-nearest neighbors. However, exact search in high dimensions is intractable due to the curse of dimensionality - the computational resources needed to solve the exact problem scale exponentially with dimensions. This has led to a diverse set of algorithms to solve the *approximate near neighbor problem*, which we now describe.

**Locality Sensitive Hashing:** LSH was the first approximate near neighbour algorithm to break the curse of dimensionality. At their core, LSH algorithms use an *LSH function* to partition the dataset into buckets. The hash function is selected so that the distance between points in the same bucket is likely to be small. To find the near neighbors of a query, we hash the query and compute the distance to every point in the corresponding bucket. Query performance can be improved with *replication*, which queries multiple independent hash tables, *multi-probe methods* [13], which examine multiple

buckets in the hash table, and *data-dependent LSH* [3], which tunes the hash function to the dataset. Recent work shows that machine learning algorithms can also construct effective LSH partitions [6].

**Count-Based LSH:** There are several recent algorithms which identify neighbors by counting the number of LSH collisions rather than explicitly computing distances. For example, the algorithm from [22] uses the count values to quickly identify potential neighbors. The algorithm from [5] applies compressed sensing techniques to the counts to compress the dataset, and a popular technique in genomics is to simply *replace* each data point with its hash values [17].

**Graphs:** Graph-based methods are another successful family of algorithms. Graph algorithms locate near neighbors by walking the edges of a graph where each point is (approximately) connected to its $k$ nearest neighbors. The focus in this area has been to improve graph properties using diversification, pruning, hierarchical structures, and other heuristics [14]. Graph indices perform well on industry-standard benchmarks but are not theoretically well-understood, despite recent progress [18]. Graph indices also suffer from long construction times and bloated memory consumption.

**Sample Compression:** A large number of practical methods are based on quantization. Such methods replace points in the dataset with *compressed versions* of the points. Methods such as scalar quantization, vector quantization and product quantization alias each point to a collection of $k$-means centroids. One can also use machine learning to obtain learned Hamming codes for the dataset and perform efficient distance computations using bit operations. Advances in quantization are applicable to most other algorithms, but have been particularly effective when combined with brute force search on GPU hardware and partition-based search over billion-scale datasets [11].

**Group Testing:** We are not the first algorithm to apply group testing to near neighbor search. However, existing algorithms have key limitations that prevent effective practical implementations and rigorous theoretical analysis. The authors of [10] propose a group-based filtering algorithm based on *group representative vectors*, or the vector average of group entries. To query the index, [10] explicitly compute the distances to all points where the distance between the representative and query exceeds a threshold. The algorithm of [20] uses the same group representatives, but applies an online backpropagation algorithm to estimate the individual similarities.

This work has two shortcomings. First, the methods require many distance calculations against the group representatives ([10] requires $N/10$ distances), resulting in poor query time. Second, the average vector can be similar to the query even when all points are far from the query, precluding a theoretical analysis except under restrictive distribution assumptions. In this work, we analyze methods where *only the group tests* are used to identify the neighbors, as our goal is to avoid performing expensive distance computations. Unsurprisingly, our method is theoretically and practically superior.

**Theoretical Comparison:** In high dimensions, most methods require conditions that depend on the data and query to prove efficient search time. Table 1 shows some representative theoretical results for the classes of algorithms described previously.

## 3 Background

**Formal Problem Statement:** In this paper, we solve the randomized *nearest* neighbor problem. Definition 1 is a stronger version of the well-studied $(R, c)$-approximate *near* neighbor problem. In particular, any algorithm which solves the randomized nearest neighbor problem also solves the approximate near neighbor problem with $c = 1$ and any $R \geq$ the distance to the nearest neighbor.

**Definition 1.** *Randomized Nearest neighbor: Given a dataset $D$, a distance metric $d(\cdot, \cdot)$ and a failure probability $\delta \in [0, 1]$, construct a data structure which, given a query point $y$, reports the point $x \in D$ with the smallest distance $d(x, y)$ with probability greater than $1 - \delta$.*

### 3.1 Group Testing

Suppose we are given a set $D$ of $N$ items, $k$ of which are positive ("hits") and $N - k$ of which are negative ("misses"). The group testing problem is to identify the hits by grouping items and using a small collection of *group tests*. A group test is positive if and only if any item from the group is positive. The objective of group testing is to reliably identify the positive items using fewer than $N$ group tests. The problem is *noisy* if the tests make i.i.d. mistakes with some false positive and false

Table 1: Asymptotic query times for several types of near neighbor algorithms. Where possible, we adapt the guarantees and notation to apply to our problem statement. For randomized algorithms, we report the time needed to succeed with constant probability - this may be boosted to an arbitrary failure rate $\delta$ in $\mathrm{polylog}(1/\delta)$ time using standard techniques.

| Method | Query Time | Notes |
|---|---|---|
| LSH [9] | $O(N^\rho \log N) \quad \rho = \frac{\log 1/s_{|K|}}{\log 1/s_{|K|+1}}$ | Many improvements to reduce $\rho$. |
| Graphs [18] | $O\left(\sqrt{d}N^{\frac{1}{d}}M^d\right)$ and $M$ is problem-dependent. | Only for dense uniform data on the unit sphere. |
| Random Trees [19] | $O\left(\frac{d \log N}{|\log \Phi|}\right) \quad \Phi(q) = \frac{1}{N-1}\sum_{x_i \neq x^\star} \frac{\mathrm{dist}(q,x_i)}{\mathrm{dist}(q,x^\star)}$ | $\Phi$ depends on the query and can be arbitrarily close to 1. |
| FLINNG [this work] | $O\left(N^{\frac{1}{2}+\gamma}\log^3(N)\right) \quad \gamma = \frac{\log s_{|K|}}{\log s_{|K|+1} - \log s_{|K|}}$ | $\gamma$ can be $< 0.5$ (query-dependent). |

negative rate. The group testing problem may also be *adaptive*, where we are allowed to design test $n$ based on the results of tests $\{1, 2, ...n-1\}$, or *non-adaptive*, where we must perform all tests at once.

Since the problem's introduction in 1943, there has been considerable work toward the construction of test designs under various constraints. For a recent review, see [1]. In this paper we develop near neighbor search algorithms using the noisy group testing framework. For simplicity, we mainly consider the *doubly regular design*, where we evenly distribute items among $B$ tests, and we independently repeat this process $R$ times to obtain a $B \times R$ grid of group tests (Figure 1). This block testing design is similar to that of RAMBO [8], but with different filtering and inference algorithms. However, our algorithmic framework is compatible with any non-adaptive testing design.

### 3.2 Locality-Sensitive Hashing

A hash function $h(x) \mapsto \{1, ..., R\}$ is a function that maps an input $x$ to an integer in the range $[1, R]$. An LSH family $\mathcal{H}$ is a set of hash functions with the following property: Under the hash mapping, nearby points have a high probability of having the same hash value. The two points $x$ and $y$ are said to *collide* if $h(x) = h(y)$. We will use the notation $s(x, y)$ to refer to the collision probability $\mathrm{Pr}_{\mathcal{H}}[h(x) = h(y)]$. The original definition of LSH given by [9] establishes lower bounds on $s(x, y)$ when $d(x, y)$ is small (i.e. we want a high probability that $x$ and $y$ collide) and upper bounds when $d(x, y)$ is large (i.e. we do not want $x$ and $y$ to collide). For our analysis, we will assume a slightly different notion of LSH. Specifically, we suppose that $s(x, y)$ is exactly equal to the similarity between $x$ and $y$. That is, $s(x, y) = \mathrm{sim}(x, y)$. The vast majority of LSH functions in the literature satisfy this property - see [7] for a review.

We also introduce the concatenation trick. For any positive integer $L$, we may transform an LSH family $\mathcal{H}$ with collision probability $s(x, y)$ into a new family having $s(x, y)^L$ by sampling $L$ hash functions from $\mathcal{H}$ and concatenating the values to obtain a new hash code $[h_1(x), h_2(x), ..., h_L(x)]$. If the original hash family had the range $[1, R]$, the new hash family has the range $[1, R^L]$.

### 3.3 Distance-Sensitive Bloom Filters

The distance-sensitive Bloom filter [12] is a data structure which solves the *approximate set membership problem*.

**Definition 2.** *Approximate Set Membership: Given a set $D$ of $N$ points and similarity thresholds $S_L$ and $S_H$, construct a data structure which, given a query point $y$, has:*
**True Positive Rate:** *If there is $x \in D$ with $\mathrm{sim}(x, y) > S_H$, the structure returns true w.p. $\geq p$*
**False Positive Rate:** *If there is no $x \in D$ with $\mathrm{sim}(x, y) > S_L$, the structure returns true w.p. $\leq q$*

The distance-sensitive Bloom filter solves this problem using LSH functions and a 2D bit array. The structure consists of $m$ binary arrays that are each indexed by an LSH function. There are three

parameters: the number of arrays $m$, a positive threshold $t \leq m$, and the number of concatenated hash functions $L$ used within each array. The length of each array is set to be the range of the LSH family and is therefore not a parameter.

To construct the filter, we insert elements $x \in D$ by setting the bit located at array index $[m, h_m(x)]$ to 1. To query the filter, we determine the $m$ hash values of the query $y$. If at least $t$ of the corresponding bits are set, we return true. Otherwise, we return false. For our group testing analysis, we need explicit bounds on the error rates $p$ and $q$. We obtain these bounds using a straightforward extension of Proposition 2.1 from [12] and provide a proof in the supplementary materials.

**Theorem 1.** *Assuming the existence of an LSH family with collision probability $s(x, y) = \text{sim}(x, y)$, the distance-sensitive Bloom filter solves the approximate membership query problem with*

$$p \geq 1 - \exp\left(-2m\left(-t + S_H^L\right)^2\right) \qquad q \leq \exp\left(-2m\left(-t + NS_L^L\right)^2\right) \tag{1}$$

---

**Algorithm 1** Index Construction

**Input:** Dataset $D$ of size $N$, positive integers $B$ and $R$, similarity threshold $S$
**Output:** A FLINNG search index consisting of membership sets $M_{r,b}$ and group tests $C_{r,b}$
**for** $r = 0$ **to** $R - 1$ **do**
   Let $\pi(D)$ be a random permutation of $D$
   Define $M_{r,b} = \{\pi(D)_i \mid i \mod B = b\}$
**for** $r = 0$ **to** $R - 1$ **do**
   **for** $b = 0$ **to** $B - 1$ **do**
      Construct a classifier $C_{r,b}$ for membership set $M_{r,b}$ with true positive rate $p$ and false positive rate $q$

**Algorithm 2** Index Query

**Input:** A FLINNG index and a query $y$
**Output:** Approximate set $\hat{K}$ of neighbors with similarity greater than the threshold $S$
$\hat{K} = \{1, \ldots, N\}$
**for** $r = 0$ **to** $R - 1$ **do**
   $Y = \emptyset$
   **for** $b = 0$ **to** $B - 1$ **do**
      **if** $C_{r,b}(y) = 1$ **then**
         $Y = Y \cup M_{r,b}$
   $\hat{K} = \hat{K} \cap Y$

---

## 4  Algorithm

We will now describe our algorithm for high-dimensional near neighbor search. We begin by reducing the near neighbor search problem to a group testing problem. Suppose we are given an $N$-point dataset $D$ and are asked to return points which are similar to a query $y$. If we apply a similarity threshold to the dataset, we obtain a near neighbor set $K = \{x \in D \mid \text{sim}(x, y) \geq S\}$. We consider $K$ to be the set of "positives" in the group testing problem. We can solve the similarity search problem by finding the $|K|$ positives among the $N - |K|$ negatives using group testing.

In order to do so, we split the dataset $D$ into a set of groups, which we visualize as a $B \times R$ grid of cells. Each cell has a group of items $M_{r,b}$ and a corresponding group test $C_{r,b}$. To assign items to cells, we evenly distribute the $N$ points among the $B$ cells in each column of the grid, and we independently repeat this assignment process $R$ times.

To identify groups that contain positives, we need a testing procedure that outputs "true" when the group contains a point similar to $y$ and "false" otherwise. That is, we require a binary classifier $C_{r,b}$ that solves the approximate membership testing problem for $M_{r,b}$ (Definition 2). For group testing to be effective, the true positive rate $p$ and false positive rate $q$ of the classifier $C_{r,b}$ should be good enough to reliably identify positive and negative cells, respectively.

Algorithm 1 shows how to construct the index. We begin by randomly distributing the points across the $B$ cells in each row, so that each cell has the same number of points. This can be done by randomly permuting the elements of $D$ and assigning blocks of $\frac{N}{B}$ elements to each cell using modulo hashing. Then, we construct classifiers (group tests) to solve the approximate membership problem in each cell.

To query the index with a point $y$, we begin by querying each classifier. If $C_{r,b}(y) = 1$, then at least one of the points in $M_{r,b}$ has high similarity to $y$. We collect all of these "candidate points" by taking the union of the $M_{r,b}$ sets for which $C_{r,b}(y) = 1$. We repeat this process for each of the $R$ repetitions

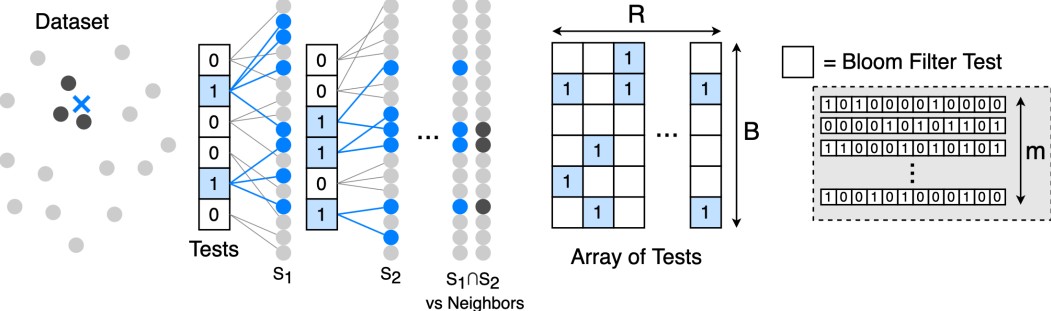

Figure 1: Intuition behind our approach. We mark the neighbors (black dots) of a query (blue X) as "positives" for group testing and construct a $B \times R$ array of tests. Based on the test results, we obtain sets $S_1, S_2, ...S_R$ of candidates, which we intersect to identify the neighbors. In general, the tests can be any classifier that detects neighbors. In this paper, we use distance-sensitive Bloom filters.

to obtain $R$ candidate sets, one for each column in the grid. With high probability, each candidate set contains the true neighbors, but it may also have some non-neighbors that were included in $M_{r,b}$ by chance. To filter out these points, we intersect the candidate sets to obtain our approximate near neighbor set $\hat{K}$. Algorithm 2 explains this process in greater detail.

**Intuition:** In each repetition, we partition $N$ points into $B$ groups, where $B \ll N$. To understand why this strategy leads to good performance, suppose we are only interested in finding the nearest neighbor $x_{\mathrm{NN}}$ (i.e. $|K| = 1$). If the tests have a high true positive rate $p$, then the $R$ cells that contain $x_{\mathrm{NN}}$ will have $C_{r,b}(y) = 1$. If the tests have a low false positive rate $q$, then the cells that do not contain $x_{\mathrm{NN}}$ will have $C_{r,b}(y) = 0$ with high probability.

In the first repetition, our tests identify the group $M_{0,b}$ that contains $x_{\mathrm{NN}}$ - each point in $M_{0,b}$ is a near-neighbor candidate. Thus, with only $B$ calls to the classifier $C_{r,b}(y)$, we have reduced the number of candidates from $N$ to $\frac{N}{B}$. If we repeat this process, we find another candidate set $M_{1,b}$. Our overall set of candidates is now the intersection $M_{0,b} \cap M_{1,b}$, whose expected size is $\frac{N}{B^2}$. In general, each repetition reduces the number of candidates by a factor of $\frac{1}{B}$, which decreases *exponentially* with the number of repetitions. We progressively rule out more and more candidates until we are left with only the near neighbors.

In practice, this process is efficient because we can construct tests with a reasonable $p$ and $q$ that are very fast to query. For example, when $C_{r,b}$ is a distance-sensitive Bloom filter, the testing process can be implemented using constant memory with bit operations or efficient integer lookup tables. The set union and intersection operations can also be implemented using cheap integer operations. The result is an algorithm that identifies the near neighbors using group testing, without explicitly storing the data or performing any distance computations.

## 5 Theory

*Proof Sketch:* We defer full proofs to the supplementary materials; what follows is a high-level description of our theory. To obtain theoretical guarantees, we first assume that the tests have a universal fixed false positive and false negative rate. Under this assumption, we derive bounds on the query time and error rates of the FLINNG algorithm. Next, we show how to construct distance-sensitive Bloom filters with a given error rate that satisfies these conditions. Here, the main technical difficulty is to bound the query time necessary to achieve the correct testing error rate. To do this, we introduce a data-dependent sparsity measure $\gamma$ that is small when the query has only a few close neighbors. To prove our main theorem, we set the test error rates so that the group design solves the randomized nearest neighbor problem.

### 5.1 Group Testing: Runtime and Accuracy

We first derive bounds on the error rates of our index. To find the probability of correctly reporting points as neighbors, we assume that each group test has a fixed false positive and negative rate.

**Lemma 1.** *Suppose we have a dataset $D$ of points, where a subset $K \subseteq D$ is "positive" and the rest are "negative." Construct a $B \times R$ grid of tests, where each test has i.i.d. false positive rate $p$ and false negative rate $q$. Then Algorithm 2 reports points as "positive" with probability:*

$$\Pr[\text{Report } x | x \in K] \geq p^R \tag{2}$$

$$\Pr[\text{Report } x | x \notin K] \leq \left[ q \left( \frac{eN(B-1)}{B(N-1)} \right)^{|K|} + p \left( 1 - \left( \frac{N(B-1)}{eB(N-1)} \right)^{|K|} \right) \right]^R \tag{3}$$

The cost of group testing inference (Algorithm 2) is the cost to do all $B \times R$ tests, plus the cost of intersecting the positive groups. We bound this cost by bounding the number of positive groups.

**Theorem 2.** *Under the assumptions in Lemma 1, suppose that each test runs in time $O(T)$. Then with probability $1 - \delta$*

$$t_{query} = O\left( BRT + \frac{RN}{B}(p|K| + qB) \log(1/\delta) \log N \right) \tag{4}$$

## 5.2 Bounding the Test Cost

Our next step is to find the runtime and error rates of a specific binary classifier: a distance-sensitive Bloom filter. To distinguish between the $K$ nearest neighbors and the rest of the dataset, we construct a filter using Theorem 1 with $S_H = \text{sim}(x_{|K|}, y) = s_{|K|}$ and $S_L = \text{sim}(x_{|K|+1}, y) = s_{|K|+1}$, where $x_{|K|}$ is $y$'s $K$th nearest neighbor. We also assume that the filter contains $\frac{N}{B}$ points and $B = 2\sqrt{N}$. Our goal in this section is to select a threshold $t$, number of bit arrays $m$, and LSH parameter $L$ to obtain a specified value of $p$ and $q$. Once we have a test with the required error rates, we will bound the test time $T$.

Without imposing additional requirements on the query and dataset, it is impossible to design a filter for an arbitrary $p$ and $q$, as observed by [12]. However, this limitation is not serious. We can obtain the error rates provided that the query has $K$ clearly-defined neighbors and the non-neighbor points are easily distinguished from the neighbors (i.e. $s_{|K|+1} \ll s_{|K|}$). This is closely related to the stability condition from [4], so we refer to such queries as *stable*. We formally define a $\gamma$-stable query as:

**Definition 3.** *$\gamma$-stable Query:* We say that a query is $\gamma$-stable if $\frac{\log(s_{|K|})}{\log(s_{|K|+1}) - \log(s_{|K|})} \leq \gamma$

We are now ready to design the classifier. Our classifier achieves the error rates $p$ and $q$ for any $\gamma$-stable query and has bounded query time, fulfilling the conditions from Theorem 2.

**Theorem 3.** *Given a true positive rate $p$, false positive rate $q$ and stability parameter $\gamma$, it is possible to choose $m$, $L$ and $t$ so that the resulting distance-sensitive Bloom filter has false positive rate $p$ and false negative rate $q$ for all $\gamma$-stable queries. The query time is*

$$O(mL) = O(-\log(\min(q, 1-p))N^\gamma \log(N)) \tag{5}$$

## 5.3 Query Time Analysis

In this section, we initialize the group testing framework using Bloom filter tests to solve the randomized nearest neighbor problem. First, we consider the query time of a $2\sqrt{N} \times R$ grid of Bloom filter classifiers. Lemma 2 is a straightforward substitution of the query time from Theorem 3 into Theorem 2, with $T = mL$.

**Lemma 2.** *Under the assumptions in Lemma 1, we can use distance-sensitive Bloom filters as tests to achieve the following query time $t_{query}$ of Algorithm 2 with probability $1 - \delta$*

$$t_{query} = O(RN^{\frac{1}{2}+\gamma} \log(N) \max(-\log(q), -\log(1-p)) + RN^{\frac{1}{2}} \log^2(N)(|K| + qN^{\frac{1}{2}}) \log(1/\delta))$$

There are two ways that Algorithm 2 can fail to solve the nearest neighbor problem (i.e. $|K| = 1$). We may fail to return the nearest neighbor, but we may also return any point in $D$ that is not the nearest neighbor. We can determine the values of $p$ and $q$ needed to achieve an overall failure rate $\delta$ by requiring that both events occur with probability $< \frac{\delta}{2}$ and applying the union bound.

**Lemma 3.** *Under the assumptions in Lemma 1, we can build a data structure that solves the randomized nearest neighbor problem for sufficiently large $N$ and small $\delta$, where*[2]

$$p = 1 - \frac{\delta}{2R} \qquad q = N^{-\frac{1}{2}} \qquad R = \frac{\log(\frac{1}{\delta})}{\log(4.80N^{\frac{1}{2}}) - \log(2e^2 + 3.44N^{\frac{1}{2}})} \tag{6}$$

We obtain our main theorem by using the values from Lemma 3 with the query time from Lemma 2. Note that the query time is sublinear when the (data-dependent) stability parameter $\gamma < \frac{1}{2}$.

**Theorem 4.** *(Main Theorem) Under the assumptions of Lemma 3, we solve the randomized nearest neighbor problem for $\gamma$-stable queries in time $t_{query}$:*

$$t_{query} = O\left(N^{\frac{1}{2}+\gamma}\log^4(N)\log^3\left(\frac{1}{\delta}\right)\right) \tag{7}$$

**Practical Implications:** We must have $s_{|K|}^{1+\gamma} < s_{|K|+1}^{\gamma}$ to have sub-linear query time ($\gamma < 0.5$). In practice, this means that the $|K|$ neighboring points should have high similarity to the query and the non-neighboring points should be far away. The high-dimensional Gaussian mixture model is a simple example of a favorable data distribution, where queries are $\gamma$-stable if the Gaussians are sufficiently well-spaced to produce tight clusters. Real datasets also happen to satisfy the $\gamma$-stable condition for many queries. For example, 15% of the queries for the RefSeqG dataset satisfy $\gamma < 0.5$ with $K = 1$. If we consider $1 \leq K \leq 100$, 35% of queries satisfy the sub-linearity condition for some $K$ in this range. We provide additional results in the supplementary materials.

Finally, the size of the index is $O(NmBR) = O(N^{\frac{3}{2}}\log^2(N))$, leading to two concerns. First, the space is asymptotically super-linear. However, the practical index size scales much better than expected because the Bloom filter space overhead involves very small constant factors (often a few bits). This leads to practical index sizes that are much smaller than other methods - even those with superior asymptotic behavior. Second, the size depends on the range $m$ of the LSH function $h(x)$. While some LSH families have small constant $m$, others have infinite $m$. However, we may simply use a univeral hash function to *rehash* the infinite output space to range $m$ in such cases. The penalty is a minor distortion to the collision probability that does not substantially affect our results.

---

**Algorithm 3** Threshold Relaxation Algorithm

---

**Input:** $A$: Array of cells, sorted in descending order by hash collisions, $k$: number of neighbors to return

**Output:** Approximate $k$ neighbors of the query
Counts $\leftarrow$ Array of length $N$, initialized to 0
Result $\leftarrow$ Empty list of IDs
**for** $i = 0$ **to** $B \times R - 1$ **do**
  **for point** $x \in$ **cell** $A[i]$ **do**
    increment Counts$[x]$
    **if** Counts$[x] = R$ **then**
      **append** $x$ **to** Result
      **if** $|$Result$| = k$ **then**
        **return** Result

---

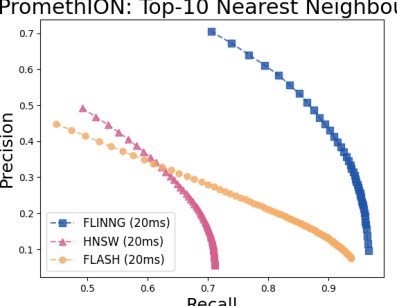

Figure 2: Precision recall tradeoff on PromethION with a query time limit of 20 ms. Up and to the right is better.

# 6 Implementation

Several nontrivial implementation tricks are needed to achieve good practical performance. First, we use the same $m$ LSH functions for all of the filters, allowing us to hash the query only one time. Second, we represent the distance-sensitive Bloom filters as lists of hash codes rather than bit arrays. This allows us to represent the FLINNG structure as a *reverse index* from hash values

---

[2]We require $N \geq 150$ and $\delta$ small enough that $R \geq 10\log N$

to cells. The reverse index is a lookup table that, given a hash value $h$, returns a list of cells whose distance-sensitive Bloom filters contain $h$. We keep a reverse index for each of the $m$ LSH functions.

To query the index, we use the reverse index to count the number of times that each cell collides with the query across the $m$ LSH functions. This results in an array of $B \times R$ count values, one for each cell. To obtain the classifier outputs, we mark all cells with count values larger than a threshold $t$ as "true." In Theorem 3, we used a global value of $t$ for all queries. However, this does not work in practice because different queries require different similarity thresholds. To address this issue, we use Algorithm 3, which relaxes $t$ until enough cells return "true" so that $k$ neighbors are returned. This process is equivalent to running Algorithm 2 with decreasing thresholds until $k$ points are returned.

We implemented FLINNG in C++, compiled with the highest level of optimization with GCC, and used OpenMP to parallelize index construction. We implemented the reverse index as an $m \times 2^L$ table of pointers to vectors, where each vector contains a list of cells. To reduce index space, we store cell identifiers as short integers when possible. Finally, note that Algorithm 3 requires an array of length $N$ to store the count values associated with each point. For large datasets, this can exceed the CPU cache size leading to a slowdown from RAM access. However, we can avoid this issue if $R = 2$ (which is often sufficient for many applications). By storing the counts as a bit array of $N/8$ bytes, we can fit $N \leq 240$ million into a 30 MB CPU cache. We make this substitution where appropriate.

## 7 Experiments

**Datasets:** We tested FLINNG on high-dimensional genomics, web-scale data mining, and embedding search datasets. We list the datasets in Table 2 and briefly describe them here. RefSeqG and RefSeqP are sets of reference genome and proteome sequences for approximately 88k species [17]. The sequences are represented as sets of 21-grams ($k$-mers) and are compressed via MinHash. Similarity search is relevant to RefSeq because we can answer basic scientific questions by clustering genomes. PromethION is a stream of raw metagenomic sequence reads from the latest sequencing machine by Oxford Nanopore [16], which generates 4TB of data per day. We preprocess the reads into 16-grams. Here, similarity search is important for read de-duplication and other pre-assembly applications. The URL dataset and Webspam datasets are from the libsvm repository. The YFCC100M dataset consists of embeddings derived from neural network activations for 100M videos and images [21].

| Dataset | $N$ | $d$ | $\bar{d}$ | Description |
|---------|-----|-----|-----------|-------------|
| RefSeqG | 117k | 1.4T | 1k | Compressed genomes |
| RefSeqP | 117k | 1.4T | 1k | Compressed proteins |
| PromethION | 3.7M | 4.3B | 286 | Raw sequencer data |
| URL | 2.4M | 3.2M | 116 | $n$-gram features |
| Webspam | 340k | 16.6M | 3.7k | $n$-gram features |
| YFCC100M | 97M | 4096 | 4096 | Neural embeddings |

Table 2: Datasets: We selected data from genomics, text and embedding problems. Datasets have $N$ points and $d$ dimensions, with an average $\bar{d}$ nonzero entries per point.

| | Memory | Indexing |
|---|--------|----------|
| FLINNG | 3.5 GB | 40 sec |
| FAISS | 3.7 GB | 12 hr |
| HNSW | >1 TB | >5 days |
| FLASH | 4.3 GB | 80 sec |

Table 3: Index characteristics for YFCC100M.

**Baselines:** We compare FLINNG against popular implementations of graph algorithms, LSH, and quantization-based search. FALCONN is a fast implementation of the traditional LSH algorithm that supports multi-probe LSH in various metric spaces [2]. FLASH is a recent LSH algorithm that uses aggregate LSH count statistics to avoid distance computations [22]. HNSW is a multi-level graph search algorithm with exceptional performance on industry-standard benchmarks [14]. We use the hnswlib library and extended it to work on genomic datasets. FAISS is a highly optimized quantization-based library used for billion-scale similarity search at Facebook [11]. Finally, we compare against a simple inverted index approach for sparse data when feasible, as well as our implementation (GROUPS) of the other group testing algorithm from [20].

**Experiment setup:** We construct indices in parallel but query using a single core. Due to limitations of baseline algorithms, we were unable to evaluate all algorithms on all tasks. Due to space constraints, we provide information about hyperparameters, experiment setup, and computing hardware in the supplementary materials.

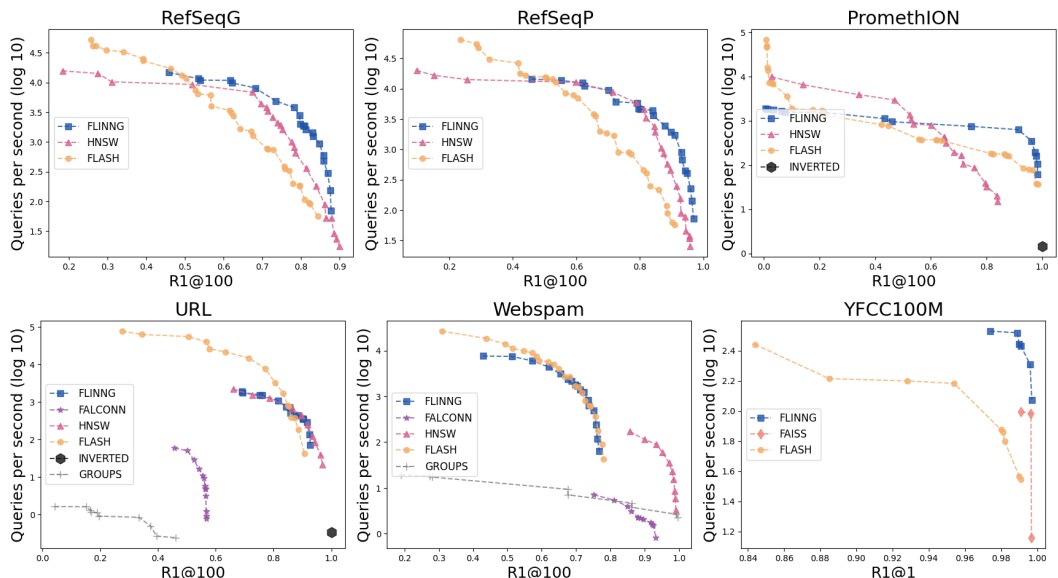

Figure 3: Time recall tradeoff for each dataset (Pareto frontier across hyperparameter configurations). Up and to the right is better. We report R1@100 except for YFCC100M, where we report R1@1.

**Results:** We show the recall-query time tradeoff for all algorithms in Figure 3. Figure 2 shows the precision-recall curve for the top 10 neighbors on the PromethION dataset when we constrain the query time to 20ms. We find that FLINNG obtains between a 2-10x speedup on many search tasks. For example, FLINNG was 3.4 times faster than FAISS at the 0.99 recall level on YFCC100M and was 4x faster than HNSW on PromethION at the 0.8 recall level. FLINNG also has a small index size and construction time when compared with baselines (Table 3).

## 8   Discussion

FLINNG is a theoretically sound algorithm with attractive practical properties that lead to a fast implementation. Our theory shows that FLINNG has sublinear query time when the query has a relatively small number of highly similar neighbors, and our experiments show that FLINNG is efficient for many real-world problems. This is particularly true in genomics, where FLINNG can outperform algorithms like HNSW by a substantial 4x margin. Existing algorithms often require expensive iterative algorithms such as $k$-means clustering or graph construction. In contrast, FLINNG relies on a simple lookup table structure that can be constructed in a single pass.

We believe that FLINNG could be particularly effective for situations where it is hard to reduce the high-dimensional similarity search to a medium-dimensional problem. In genomics, this problem is difficult because the $k$-mer distribution contains billions of items and often has a heavy-tailed distribution. The situation may also arise for embedding tasks such as YFCC100M, where it is known that dimensionality reduction can hurt performance. In these scenarios, where we prefer to search over the original metric space, FLINNG provides a fast and scalable solution.

**Limitations and Ethical Considerations:** We do not foresee any ethical problems. The main limitation of our method is that it works best on high-dimensional search tasks where the neighbors are all above a (relatively high) similarity threshold. For this reason, FLINNG is likely not the best choice for problems such as $k$-NN classification, where low-similarity results may be important.

## Acknowledgements

This work was supported by National Science Foundation IIS-1652131, BIGDATA-1838177, RI-1718478, AFOSR- YIP FA9550-18-1-0152, Amazon Research Award, and the ONR BRC grant on Randomized Numerical Linear Algebra.

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
