# Practical Near Neighbor Search via Group Testing: Supplementary Materials

**Joshua Engels**[*]
Department of Computer Science
Rice University
Houston, Texas, USA
`jae4@rice.edu`

**Benjamin Coleman**[*]
Electrical and Computer Engineering
Rice University
Houston, Texas, USA
`ben.coleman@rice.edu`

**Anshumali Shrivastava**
Department of Computer Science
Rice University & ThirdAI Corp.
Houston, Texas, USA
`anshumali@rice.edu`

## 1 Proofs

In this section, we provide proofs for all of the theorems introduced in the main text. We begin with a simple extension of the results of [3] for the Bloom filter false positive and negative rates. Then, we prove our main claim, which is that the query time of our data structure is sublinear, given some relatively weak assumptions on the stability of the query.

**Theorem 1.** *Assuming the existence of an LSH family with collision probability $s(x, y) = \text{sim}(x, y)$, the distance-sensitive Bloom filter solves the approximate membership query problem with*

$$p \geq 1 - \exp\left(-2m\left(-t/m + S_H^L\right)^2\right) \tag{1}$$

$$q \leq \exp\left(-2m\left(-t/m + NS_L^L\right)^2\right) \tag{2}$$

*Proof.* We begin with a brief explanation of the results from [3]. Recall that a distance-sensitive Bloom filter is a collection of $m$ bit arrays. Array $i$ is indexed using an independent LSH function $l_i(x)$. To insert a point $x$ into the $i^{\text{th}}$ array, we set the bit at location $l_i(x)$ to '1.' To query the filter, we calculate the $m$ hash values of the query and return "true" when at least $t$ of the corresponding bits are '1.'

To bound $p$ (the true positive rate) and $q$ (the false positive rate), we bound the probability that a single array returns "true." Since the arrays are independent, the number of '1's follows a Binomial distribution. [3] obtain their main result (Theorem 3.1, in their paper) using the Azuma-Hoeffding inequality to bound the tail of the Binomial distribution. This is done for the Hamming metric, using a specially-chosen value of $t$. We repeat their analysis, but for any value of $t$ and in a general metric space.

**True Positive Rate:** First, we will prove the bound for $p$, the true positive rate. Given a query $y$, let $p_y$ be the probability

$$p_y = \Pr[\text{bit at } h(y) = 1]$$

---

[*]Equal contribution.

35th Conference on Neural Information Processing Systems (NeurIPS 2021).

From Proposition 2.1 of [3], we have that

$$p_y \geq \max_{x \in D} \operatorname{sim}(x, y)^L > S_H^L$$

Note that $\max \operatorname{sim}(x, y) \geq S_H$ because there is a point $x \in D$ with $\operatorname{sim}(x, y) \geq S_H$, as $y$ is a true positive. Also, note that the $\frac{Nm}{\operatorname{len}}$ term (where $\operatorname{len}$ is the length of the array) is zero in our case, as we do not perform rehashing. Finally, note that we concatenate $L$ hashes so that the LSH function has collision probability $\operatorname{sim}(x, y)^L$.

We use the Azuma-Hoeffding inequality as in [3]. Let $B(q_y) = \operatorname{Binom}(m, q_y)$ and observe that

$$t - \mathbb{E}[B(q_y)] = t - mq_y \leq t - mS_H^L$$

$$
\begin{align}
1 - p &= \Pr[B(q_y) < t] \tag{3}\\
&= \Pr[B(q_y) - \mathbb{E}[B(q_y)] < t - B(q_y)] \tag{4}\\
&\leq \Pr[B(q_y) - \mathbb{E}[B(q_y)] < t - mS_H^L] \tag{5}\\
&\leq \exp\left(-\frac{2}{m}(t - mS_H^L)^2\right) \tag{6}
\end{align}
$$

Finally, we have

$$p \geq 1 - \exp\left(-2m(-t/m + S_H^L)\right)$$

**False Positive Rate:** The false positive analysis is similar. Here, we have (again, from Proposition 2.1 of [3]) that

$$p_y \leq \sum_{x \in D} \operatorname{sim}(x, y)^L \leq N \max_{x \in D} \operatorname{sim}(x, y)^L = NS_L^L$$

We will use the Azuma-Hoeffding inequality again, this time using the fact that

$$t - \mathbb{E}[B(p_y)] \geq t - mNS_L^L$$

$$
\begin{align}
q &= \Pr[B(q_y) \geq t] \tag{7}\\
&= \Pr[B(q_y) - \mathbb{E}[B(q_y)] \geq t - B(q_y)] \tag{8}\\
&\leq \Pr[B(q_y) - \mathbb{E}[B(q_y)] \geq t - mNS_L^L] \tag{9}\\
&\leq \exp\left(-\frac{2}{m}(t - mNS_L^L)^2\right) \tag{10}
\end{align}
$$

Leaving us with the desired inequality:

$$q \leq \exp\left(-2m(-t/m + S_L^L)\right)$$

$\square$

## 1.1 Group Testing: Runtime and Accuracy

The first step of our analysis is to bound the true positive rate and false positive rate of our group testing design. We suppose that the individual tests have a true positive rate $p$ and a false positive rate $q$, and that the tests are independent. Under these assumptions, we have the following result:

**Lemma 1.** *Suppose we have a dataset $D$ of points, where a subset $K \subseteq D$ is "positive" and the rest are "negative." Construct a $B \times R$ grid of tests, where each test has i.i.d. true positive rate $p$ and false positive rate $q$. Then Algorithm 2 reports points as "positive" with probability:*

$$\Pr[\text{Report } x | x \in K] \geq p^R \tag{11}$$

$$\Pr[\text{Report } x | x \notin K] \leq \left[q\left(\frac{eN(B-1)}{B(N-1)}\right)^{|K|} + p\left(1 - \left(\frac{N(B-1)}{eB(N-1)}\right)^{|K|}\right)\right]^R \tag{12}$$

*Proof.* The procedure returns "positive" for an element $x \in D$ only if all of the tests whose group contains $x$ return "positive." Therefore, we will analyze the probability of reporting $x \in D$ as "positive" in each repetition (column) of the grid. We obtain the final probabilities using the fact that the repetitions are independent.

**True Positive Rate:** If $x \in K$, then each group containing $x$ returns "positive" with the true positive rate $p$. Therefore,

$$\Pr[\text{Report } x | x \in K] = p^R$$

**False Positive Rate:** If $x \notin K$, then there are two ways for the repetition to accidentally report $x$ as "positive." The first way is for $x$ to fall into a group that contains one of the $|K|$ true positives, and for this test to correctly report "positive" (which happens with probability $p$). The second way is for $x$ to fall into a group that exclusively contains negatives, but for the test to malfunction (which occurs with probability $q$).

Let $p_x$ be the probability that we report $x$ in a single repetition. That is,

$$\Pr[\text{Report } x | x \notin K] = p_x^R$$
$$= p\Pr[x \in \text{positive group}] + q\Pr[x \in \text{negative group}]$$

The probability that $x$ falls into a negative group is determined by the hypergeometric distribution. Each group contains $N/B$ points[2] from the dataset, which we draw from a pool of $N - |K|$ negatives and $|K|$ positives. Since we condition on $x$ being negative, we draw $N/B - 1$ points from $N - 1$ possibilities, $|K|$ of which are positive. Therefore, the probability that $x$ falls into a negative group is equal to the probability mass function of the hypergeometric distribution, evaluated at zero.

$$\Pr[x \in \text{negative group}] = \frac{\binom{|K|}{0}\binom{N-|K|-1}{N/B-1}}{\binom{N-1}{N/B-1}}$$

Which, after simplification and using Vandermonde's identity:

$$\Pr[x \in \text{negative group}] = \frac{\binom{N-N/B}{|K|}}{\binom{N-1}{|K|}}$$

$$\Pr[x \in \text{positive group}] = 1 - \Pr[x \in \text{negative group}]$$

We wish to have an upper bound on both quantities, which amounts to having both an upper and lower bound on $\Pr[x \in \text{negative group}]$. We repeatedly apply the following inequalities

$$\left(\frac{a}{b}\right)^b \leq \binom{a}{b} \leq \left(\frac{ea}{b}\right)^b$$

to get

$$\frac{\left(\frac{N-N/B}{|K|}\right)^{|K|}}{\left(e\frac{N-1}{|K|}\right)^{|K|}} \leq \frac{\binom{N-N/B}{|K|}}{\binom{N-1}{|K|}} \leq \frac{\left(e\frac{N-N/B}{|K|}\right)^{|K|}}{\left(\frac{N-1}{|K|}\right)^{|K|}}$$

or, after simplification

$$\left(\frac{N(B-1)}{eB(N-1)}\right)^{|K|} \leq \frac{\binom{N-N/B}{|K|}}{\binom{N-1}{|K|}} \leq \left(\frac{eN(B-1)}{B(N-1)}\right)^{|K|}$$

This yields the inequalities

$$\Pr[x \in \text{negative group}] \leq \left(\frac{eN(B-1)}{B(N-1)}\right)^{|K|}$$

$$\Pr[x \in \text{positive group}] \leq 1 - \left(\frac{N(B-1)}{eB(N-1)}\right)^{|K|}$$

which, when substituted into the expression for $p_x^R$, proves the theorem. $\square$

---

[2]We suppose that $B$ evenly divides $N$ for simplicity. We may accommodate the general case by replacing $N$ in our inequality with $N + 1$, which does not asymptotically change our results.

We next bound the runtime of our method.

**Theorem 2.** *Under the assumptions in Lemma 1, suppose that each test runs in time $O(T)$. Then with probability $1 - \delta$*

$$t_{query} = O\left(BRT + \frac{RN}{B}(p|K| + qB)\log(1/\delta)\log N\right) \qquad (13)$$

*Proof.* We must query each group test, and then intersect all the candidate groups. The $BRT$ term is the cost of querying all $B \times R$ cells. To obtain the cost of intersecting the $R$ candidate sets, let $P_i$ be the candidate set of the $i^{\text{th}}$ repetition and let $c_i$ be the cost of the $i^{\text{th}}$ intersection, where $i = \{1, 2, ..., R\}$. The total cost is

$$c_{\text{total}} = \sum_{i=1}^{R-1} O\left(|P_{i+1}| + \left|\bigcap_{j=1}^{i} P_j\right|\right) + O\left(|P_{i=1}|\log|P_{i=1}| + \left|\bigcap_{j=1}^{i} P_j\right|\log\left|\bigcap_{j=1}^{i} P_j\right|\right)$$

because the cost to intersect two sets of sorted integers is the sum of set cardinalities, and we pay an $O(|P|\log|P|)$ cost to sort a list of size $|P|$. Also, note that

$$\left|\bigcap_{j=1}^{i} P_j\right| \leq |P_i|$$

because $|A \cap B| \leq \min\{|A|, |B|\} \leq |A|$. Therefore, we have

$$c_{\text{total}} = \sum_{i=1}^{R-1} |P_i| + |P_{i+1}| + O(N\log N) = O\left(RN\log N + \sum_{i=1}^{R} |P_i|\right)$$

Because each group has exactly $\frac{N}{B}$ points, the value of $|P_i|$ is $\sum_{j=1}^{B} \frac{N}{B}\mathbb{1}_{(i,j)}$, where the indicator function $\mathbb{1}_{(i,j)} = 1$ if the group test in row $j$ and column $i$ outputs "positive." Under the mild assumption that $p > q$ (i.e. the true positive rate is larger than the false positive rate), this sum is maximized when all $|K|$ true positives are assigned to different groups. The expected value of this sum is

$$\mu = \mathbb{E}[|P_i|] \leq p|K| + q(B - |K|) \leq p|K| + qB$$

We want to bound this sum in probability:

$$\Pr\left[\sum_{j=1}^{B} \mathbb{1}_{(i,j)} \geq (1 + \Delta)\mu\right]$$

We use the simplified Chernoff bound for independent non-identical Bernoulli sums:

$$\Pr\left[\sum_{j=1}^{B} \mathbb{1}_{(i,j)} \geq (1 + \Delta)\mu\right] \leq e^{-\frac{\Delta^2\mu}{2+\Delta}}$$

We wish to find the value of $\Delta$ which makes this probability smaller than the failure rate.

$$e^{-\frac{\Delta^2\mu}{2+\Delta}} < \delta$$

$$\frac{\Delta^2\mu}{2+\Delta} \geq \log 1/\delta$$

In our context, $\mu > 1$ (otherwise, it is trivial to bound $|P_i|$) and we may constrain $\Delta > 1$. This yields the inequality

$$\frac{\Delta^2\mu}{2+\Delta} \geq \frac{\Delta^2}{3\Delta}\log 1/\delta$$

Therefore, we may set $\Delta = 3 \log 1/\delta$ to get the following statement with probability $1 - \delta$

$$\sum_{j=1}^{B} \mathbb{1}_{(i,j)} < (1 + 3 \log(1/\delta)) \left( p|K| + qB \right)$$

This bound on the intersection cost proves the theorem.

$\square$

## 1.2 Bounding the Test Cost

In this section, we bound the runtime and error characteristics of a specific binary classifier: a distance-sensitive Bloom filter. To distinguish between the $K$ nearest neighbors and the rest of the dataset, we apply Theorem 1 to a group with $\frac{N}{B}$ points, $S_H = \text{sim}(x_{|K|}, y) = s_{|K|}$, and $S_L = \text{sim}(x_{|K|+1}, y) = s_{|K|+1}$, where $x_{|K|}$ is the $K$ nearest neighbor to the query $y$. This gives us the following bounds on $q$ and $p$:

$$p \geq 1 - exp\left(-2m\left(-\frac{t}{m} + \left(s_{|K|}\right)^L\right)^2\right) \tag{14}$$

$$q \leq \exp\left(-2m\left(-\frac{t}{m} + \frac{N}{B}\left(s_{|K|+1}\right)^L\right)^2\right) \tag{15}$$

These bounds have four design parameters that we may freely choose: $t$, the threshold number of collisions we require to report a "positive"; $m$, the number of bit arrays in the Bloom filter; $L$, the number of hash values we concatenate together in each array; and $B$, the number of cells into which the dataset is partitioned within each column. In this section, we seek to find specific values for these free parameters that will allow us to build a distance sensitive Bloom filter with sufficiently high $p$ and low $q$. We will use these values in the proof of Theorem 3 and for the rest of our analysis.

**A note about the hashing cost:** The cost to query each filter is the cost of performing $L \times m$ LSH computations. However, the LSH computations are not $O(1)$, they are $O(d)$, where $d$ is the dimensionality of the dataset. Since this simply adds a constant multiplier term of $d$ to the asymptotic expressions, we not not include the dependency on $d$ in our analysis.

**Choosing a Value For $t$:** In the bounds given in Equation (14) and Equation (15), the inner expression with $-\frac{t}{m}$ is squared, which gives the initial impression that any value of $t$ will work. However, when we derived these bounds in Theorem 1, we implicitly require that the threshold ratio (here $\frac{t}{m}$) be smaller than the "positive" Bloom filter collision probability $S_H^L$ and larger than the "negative" Bloom filter collision probability $NS_L^L$. Making the same substitutions, we find that $\frac{t}{m}$ must satisfy the following condition.

$$\frac{N}{B}(s_{|K|+1})^L < \frac{t}{m} < (s_{|K|})^L \tag{16}$$

The intuition behind this condition is that $t$ must be a threshold number of collisions that lies between the expected number of collisions in a positive group and the expected number of collisions in a negative one.

Any $\frac{t}{m}$ in this range gives us *some* valid bound on $p$ and $q$, but values very close to the edges of the range are suboptimal. Rather than find an optimal value of $t$, which is likely difficult and data-dependent, we choose a specific value for $\frac{t}{m}$ that works well: the average between the lower bound and the upper bound in Equation (16). This gives us the following value for $t$:

$$t = m\left(\frac{\frac{N}{B}(s_{|K|+1})^L + (s_{|K|})^L}{2}\right) \tag{17}$$

One benefit of choosing this value for $t$ is that the bounds on $p$ and $q$ from Equation (14) and Equations (15) now look the same. After substituting $t$ into the bounds, we define a new variable $\alpha$ as

$$\alpha = \exp\left(-2m\left(\frac{(s_{|K|})^L - \frac{N}{B}(s_{|K|+1})^L}{2}\right)^2\right) \tag{18}$$

Note that

$$q \leq \alpha \text{ and } p \geq 1 - \alpha \tag{19}$$

$\alpha$ simplifies the analysis because it represents the bounds on $p$ and $q$ at the same time. If we decrease $\alpha$, we have a larger $p$ and a smaller $q$ (i.e. a more accurate test).

**Choosing a Value For $B$ and $L$:** To further simplify the analysis, we wish to decouple $\alpha$ from $N$ in Equation (18). Our goal in this section is to choose $B$ and $L$ as a function of $N$ so that the error rate $\alpha$ no longer depends on $N$. As with $t$, our choices are not necessarily optimal. We use them because they allow us to prove theoretical guarantees about the system.

We first let

$$B = 2\sqrt{N} \tag{20}$$

To have sublinear query time, $B$ must be proportional to some fractional power of $N$ because the query time in Theorem 2 contains both $\frac{N}{B}$ and $B$ factors. The use of $N^{\frac{1}{2}}$ minimizes the complexity of their sum, and the constant factor of $2$ is chosen to simplify the analysis in the next paragraph.

We next let $L$ be the smallest positive integer such that $(s_{|K|})^L \geq 2\frac{N}{B}(s_{|K|+1})^L$. Since $s_{|K|} \geq s_{|K|+1}$, it is always possible to find such an integer (even though this integer may be impractically large). This choice of $L$ simplifies the difference in the squared term in $\alpha$ to $\frac{(s_{|K|})^L}{2}$. In particular, $\alpha$ no longer depends on $N$. The analysis now depends on $N$ exclusively through the parameter $L$.

If we start with $\frac{N}{B}(s_{K+1})^L = \frac{(s_K)^L}{2}$ and solve for L, as well as plug in our expression for $B$ from Equation (20), we get the following expression for L.

$$L = \frac{\frac{1}{2}\log(N)}{\log(s_{|K|}) - \log(s_{|K|+1})} \tag{21}$$

Finally, we can plug this value of $L$ into Equation (18) and simplify, which gives us the following value for $\alpha$.

$$\alpha = \exp\left(\frac{-m(s_{|K|})^{\frac{\log(N)}{\log(s_{|K|}) - \log(s_{|K|+1})}}}{8}\right) \tag{22}$$

We will use the following well-known fact: for any non-negative and nonzero real numbers $a, b, c$,

$$a^{\frac{\log(b)}{c}} = b^{\frac{\log(a)}{c}} \tag{23}$$

We apply Equation (23) to Equation (22), with $a = s_{|K|}$, $b = N$, and $c = \log(s_{|K|}) - \log(s_{|K|+1})$ and simplify the result to obtain

$$\alpha = \exp\left(\frac{-mN^{\frac{\log(s_K)}{\log(s_{|K|}) - \log(s_{|K|+1})}}}{8}\right) \tag{24}$$

**The $\gamma$-Stable Query Condition:**

As observed by [3], it is not possible to design a distance-sensitive Bloom filter with arbitrary $p$ and $q$ without additional conditions on the query. Therefore, we introduce the requirement that the query be $\gamma$-*stable*. That is, we require that the query not be a pathologically difficult query for a distance-sensitive Bloom filter to answer[3]. When the $\gamma$ is small, the query has $|K|$ neighbours with a clear distinction between non-neighboring points (i.e. $s_{|K|+1} \ll s_{|K|}$). When $|K|$ is large, there is no such distinction, and the neighboring points are approximately as close as the neighbors. Unstable queries are both rare and non-informative in the near neighbor setting - for further discussion, see the seminal paper of [2]. We formally define a $\gamma$-stable query as:

**Definition 3. $\gamma$-*stable Query*:** *We say that a query is $\gamma$-stable if*

$$\frac{\log(s_{|K|})}{\log(s_{|K|+1}) - \log(s_{|K|})} \leq \gamma \tag{25}$$

---

[3]This parameter functions similarly to $\rho$ from the standard LSH near neighbor analysis.

First we note that similarity is a measure from $0$ to $1$, so the numerator of $\gamma$ is negative. Furthermore, since $s_{|K|+1} < s_{|K|}$ and the $\log$ function is monotonically increasing, $s_{|K|+1} - s_{|K|} < 0$ and the denominator is negative. Thus, $\gamma$ is positive (a negative number over a negative number). Indeed, if $\gamma$ is small then the query is stable because $\gamma$ is small when the similarity of $x$ with the $|K|^{\text{th}}$ nearest neighbor is large (the numerator is a small negative number) and when there is a large similarity gap between the $|K|^{\text{th}}$ and $|K+1|^{\text{th}}$ neighbors (the denominator is a large negative number).

Our $\gamma$-stable parameterization has the added benefit of removing the similarity values $s_{|K|}$ and $s_{|K|+1}$ from Equation (24). Substituting the query parameterization from Definition 3 to Equation (24), we have the following upper bound for $\alpha$. This bound holds for all $\gamma$-stable queries:

$$\alpha \leq \exp\left(\frac{-mN^{-\gamma}}{8}\right) \tag{26}$$

The reason Equation (26) is an upper bound is because Definition 3 guarantees that $-\gamma$ is a *larger in magnitude* negative number than the negative exponent of $N$ in Equation (24). Thus, when we raise $N$ to these exponents, we have that

$$N^{-\gamma} < N^{\frac{\log(s_{|K|})}{\log(s_{|K|}) - \log(s_{|K|+1})}} \tag{27}$$

Thus, our use of $\gamma$ results in a *smaller* (in magnitude) negative number inside the exponential in Equation (24), and thus a larger $\alpha$ overall. Note that this somewhat loosens our bounds for $p$ and $q$ (since they are in terms of $\alpha$).

Given these parameter choices, we are ready to state our theorem which bounds the query time of each distance-sensitive Bloom filter. We use $p'$ and $q'$ to denote the desired true positive rate and false positive rate of the Bloom filter, to differentiate from $p$ and $q$ above.

**Theorem 3.** *Given a true positive rate $p'$, false positive rate $q'$ and stability parameter $\gamma$, it is possible to choose $m$, $L$ and $t$ so that the resulting distance-sensitive Bloom filter has true positive rate $p'$ and false positive rate $q'$ for all $\gamma$-stable queries. The query time is*

$$O(mL) = O\left(-\log(\min(q', 1 - p'))N^{\gamma}\log(N)\right) \tag{28}$$

*Proof.* From Equation (19), we have $q \leq \alpha$ and $p \geq 1 - \alpha$. To guarantee that the actual error rates of the filter meet the design requirements (i.e. $q \leq q'$ and $p \geq p'$), we must choose $\alpha$ small enough such that $q' \leq \alpha$ and $p' \geq 1 - \alpha$. Thus we need

$$\alpha \leq \min(q', 1 - p') \tag{29}$$

For $\gamma$-stable queries, we may bound $\alpha$ using Equation (26). Because it is more expensive to design a filter with small $\alpha$, we wish to use the largest possible value of $\alpha$ that will work. This value is attained when the upper bound in Equation (26) is equal to the upper bound in Equation (29). This gives us the following condition for the Bloom filter to have the desired error characteristics:

$$\exp\left(\frac{-mN^{-\gamma}}{8}\right) = \min(q, 1 - p) \tag{30}$$

Simplifying, we have the following expression for $m$. Note that $m$ is positive because $\log(\min(q, 1 - p)) < 0$.

$$m = -8\log(\min(q, 1 - p))N^{\gamma} \tag{31}$$

We now have concrete values for all of our free variables: $t$ from Equation (17), $B$ from Equation (20), $L$ from Equation (21), and $m$ from Equation (31). Note that $L$ should be chosen as the maximum over all $s_{|K|}$ and $s_{|K|+1}$ that are $\gamma$-stable[4], so that Equation (21) is true for *all* $\gamma$-stable queries.

When $t$,$B$,$L$, and $m$ are chosen in this way, the resulting filter has true positive rate $p \leq p'$ and false positive rate $q \leq q'$ for all $\gamma$-stable queries.

To query a distance-sensitive Bloom filter, we must compute $m \times L$ hash functions and perform $m$ array lookups for an overall complexity of $O(mL)$. Thus, the time to query our Bloom filter is

$$O(mL) = O\left(\log(\min(q, 1 - p))N^{\gamma}\log(N)\right)$$

$\square$

---

[4]This requires a mild assumption that the ratio $\frac{s_{|K|}}{s_{|K|+1}}$ is bounded.

## 1.3 Query Time Analysis

In this section, we combine previous results to solve the randomized nearest neighbor problem. First, we consider the query time of a $2\sqrt{N} \times R$ grid of Bloom filter classifiers. Lemma 2 is a straightforward substitution of the query time from Theorem 3 into Theorem 2, with $T = mL$.

**Lemma 2.** *Under the assumptions in Lemma 1, we can use distance-sensitive Bloom filters as tests to achieve the following query time $t_{query}$ of Algorithm 2 with probability $1 - \delta$*

$$t_{query} = O(RN^{\frac{1}{2}+\gamma}\log(N)\max(-\log(q), -\log(1-p)) \\ + RN^{\frac{1}{2}}\log^2(N)(|K| + qN^{\frac{1}{2}})\log(1/\delta)) \tag{32}$$

*Proof.* Each cell is a distance-sensitive Bloom filter, so the test query time $T$ is equal to our result for the query time of a distance-sensitive Bloom filter from Theorem 3.

We use $B = 2\sqrt{N}$ from Equation (20) and $O(T) = O\left(\log(\min(q, 1-p))N^\gamma \log(N)\right)$ from Theorem 3 with the query time expression from Theorem 2 to get

$$t_{query} = O(-RN^{\frac{1}{2}+\gamma}\log(N)\log(\min(q, 1-p)) \\ + RN^{\frac{1}{2}}\log^2(N)(p|K| + qN^{\frac{1}{2}})\log(1/\delta)) \tag{33}$$

Since $p < 1$, we may replace it with 1. Also, since $0 < p < 1$ and $0 < q < 1$,

$$-\log(\min(p, q)) = \max(-\log(p), -\log(q)) \tag{34}$$

Making these two substitutions into Equation (33), we have

$$t_{query} = O(RN^{\frac{1}{2}+\gamma}\log(N)\max(-\log(q), -\log(1-p)) \\ + RN^{\frac{1}{2}}\log^2(N)(|K| + qN^{\frac{1}{2}})\log(1/\delta)) \tag{35}$$

$\square$

Our bound on the query time has two free parameters: $p$ and $q$. We will show that, given an appropriate choice for $p$ and $q$, our algorithm solves the nearest neighbor problem (i.e. $|K| = 1$) in sublinear time. We begin by finding the values of $p$ and $q$ which solve the nearest neighbor problem in Lemma 3.

**Lemma 3.** *Under the conditions in Lemma 1, our data structure solves the randomized nearest neighbor problem for sufficiently large $N$ and small $\delta$, where[5]*

$$p = 1 - \frac{\delta}{2R} \qquad q = N^{-\frac{1}{2}} \tag{36}$$

$$R = \frac{\log(\frac{1}{\delta})}{\log(4.80N^{\frac{1}{2}}) - \log(2e^2 + 3.44N^{\frac{1}{2}})} \tag{37}$$

*Proof.* For this proof we will use the index described in Algorithm 1, using $R$ columns of $B = 2\sqrt{N}$ distance-sensitive Bloom filter cells. We now will determine the requirements for $p$, $q$, and $R$ to achieve an overall failure rate of $\delta$ in Algorithm 2.

There are two ways that the querying process, Algorithm 2, can fail to solve the nearest neighbor problem. We may fail to return the nearest neighbor, but we may also return any point in $D$ that is not the nearest neighbor. Let $P$ be the probability that Algorithm 2 returns the nearest neighbor $x_{NN}$ to the query, and let $Q$ be the probability Algorithm 2 returns at least one other point in $D$. Then using the union bound, we solve the nearest neighbor problem if

$$(1 - P) + Q < \delta \tag{38}$$

To simplify our analysis, we constrain $(1 - P)$ and $Q$ to be less than $\frac{\delta}{2}$.

$$(1 - P) \leq \frac{\delta}{2} \qquad Q < \frac{\delta}{2} \tag{39}$$

---

[5]We require $N \geq 150$ and $\delta$ small enough that $R \geq 10 \log N$

**Analysis of** $1 - P$**:** Since $|K| = 1$ for the nearest neighbor problem, the true positive rate from Theorem 1 is equal to $P$, so that $P \geq p^R$. Thus $1 - P \leq 1 - p^R$, so we will achieve the necessary bound on $P$ in Equation 39 if

$$1 - p^R \leq \frac{\delta}{2} \tag{40}$$

Rearranging the inequality in terms of $1 - p$, we have a constraint on the values of $p$ which solve the nearest neighbor problem.

$$1 - p \leq 1 - \left(1 - \frac{\delta}{2}\right)^{\frac{1}{R}} \tag{41}$$

If $p$ satisfies the inequality, then $1 - P < \frac{\delta}{2}$). Thus, we may *reduce* the right hand side of the inequality, and the resulting values of $p$ will *still* solve the nearest neighbor problem.

We now tighten the constraint in Equation (41). We use the Generalized Bernoulli's inequality, which states that

$$(1 - x)^r \leq 1 - rx \qquad \text{for } r \in [0, 1] \tag{42}$$

Since $R \geq 1$, $\frac{1}{R} \in [0, 1]$, so we can apply this to the right side of Equation (41):

$$1 - \left(1 - \frac{\delta}{2}\right)^{\frac{1}{R}} \geq \frac{\delta}{2R} \tag{43}$$

This gives us our new, more restrictive constraint for $1 - p$:

$$1 - p \leq \frac{\delta}{2R} \tag{44}$$

**Analysis of** $Q$**:** From Theorem 1, we have an upper bound on the probability that a single point is falsely reported, $\Pr[\text{Report } x | x \notin K]$. Using the union bound, we get that $Q$, the probability that *any* of the $N$ points is falsely reported, is less than or equal to $N$ times the probability that a single point is falsely reported:

$$Q \leq N * \Pr[\text{Report } x | x \notin K]^N \tag{45}$$

Thus, we can achieve the requirement from Equation (39) that $Q \leq \frac{\delta}{2}$ by requiring that

$$N * \Pr[\text{Report } x | x \notin K]^N < \frac{\delta}{2} \tag{46}$$

If we now substitute in our expression for $\Pr[\text{Report } x | x \notin K]$ with $|K| = 1$ and $B = 2\sqrt{N}$ from Theorem 1 and (extensively) simplify, we have a constraint for $q$.

$$q < \frac{2eN^{\frac{1}{2}}}{e^2(2N^{\frac{1}{2}} - 1)} \left[ \frac{N-1}{N} \left(\frac{\delta}{2N}\right)^{\frac{1}{R}} - \frac{p[(2e\frac{N-1}{N} - 2)N^{\frac{1}{2}} + 1]}{2eN^{\frac{1}{2}}} \right] \tag{47}$$

Like we did above for $1 - p$, we can now tighten this constraint for $q$ to obtain a simpler expression. The simplified constraint leads to a smaller range of values for $q$, but these values still satisfy the original constraint and guarantee a total error rate of $\delta$. We decrease the constraint by replacing the factor of $-p$ with $-1$, by replacing the factor of $-\frac{N-1}{N}$ with $-1$, and by replacing the factor of $\frac{e(2N^{\frac{1}{2}})}{e(2N^{\frac{1}{2}}-1)}$ with 1. We end up with the following (tighter) constraint for $q$:

$$q < \frac{N-1}{Ne} \left(\frac{\delta}{2N}\right)^{\frac{1}{R}} - \frac{(2e-2)N^{\frac{1}{2}} + 1}{2e^2 N^{\frac{1}{2}}} \tag{48}$$

Breaking up the $\frac{\delta}{2N}$ term and simplifying, we get

$$q < \left(\frac{N-1}{N}\right) \frac{2^{1-\frac{1}{R}} N^{\frac{1}{2} - \frac{1}{R}} \delta^{\frac{1}{R}}}{2eN^{\frac{1}{2}}} - \frac{(2e-2)N^{\frac{1}{2}} + 1}{2e^2 N^{\frac{1}{2}}} \tag{49}$$

Note that as $R$ increases, the right hand side of the constraint for $q$ also increases. For some small values of $R$, the right hand side is actually negative. A negative expression means that we have shrunk the range of allowable $q$ values so much that our simplified constraint is no longer meaningful. This does not mean that it is impossible to find $q$ to satisfy the original constraint, it simply means that our simplifications were too aggressive.

We now make two key assumptions that allow us to show that the right hand side of Equation (48) is always positive and well defined:

$$N \geq 150 \qquad R \geq 10 \log N > 50 \tag{50}$$

The analysis is possible without these assumptions, but must be done with the complicated expression in Equation 47 rather than the simple version.

We continue to tighten the constraint by replacing some values of $N$ and $R$ with their smallest possible values (i.e. $N = 150$ and $R = 50$), in cases where making such a replacement will only make the right hand side of the constraint smaller[6]:

$$q < \frac{149}{150} \frac{2^{\frac{49}{50}} N^{\frac{1}{2} - \frac{1}{10 \log N}} \delta^{\frac{1}{R}}}{2e N^{\frac{1}{2}}} - \frac{(2e - 2)N^{\frac{1}{2}} + 1}{2e^2 N^{\frac{1}{2}}} \tag{51}$$

We also simplify the term $N^{\frac{1}{2} - \frac{1}{10 \log N}}$:

$$N^{\frac{1}{2} - \frac{1}{10 \log N}} = N^{\frac{1}{2}} N^{\frac{-1}{10 \log N}}$$
$$= N^{\frac{1}{2}} \left( N^{\frac{1}{\log N}} \right)^{\frac{-1}{10}}$$
$$= N^{\frac{1}{2}} e^{\frac{-1}{10}} \qquad\qquad \text{since } x^{\frac{1}{\log x}} = e$$

Plugging this value back into the constraint, we have

$$q < \frac{\frac{149}{150} 2^{\frac{49}{50}} N^{\frac{1}{2}} e^{\frac{-1}{10}} \delta^{\frac{1}{R}}}{2e N^{\frac{1}{2}}} - \frac{(2e - 2)N^{\frac{1}{2}} + 1}{2e^2 N^{\frac{1}{2}}} \tag{52}$$

To obtain our final constraint for $q$, we first combine fractions by multiplying the top and bottom of the left fraction by $e$:

$$q < \frac{\frac{149}{150} 2^{\frac{49}{50}} N^{\frac{1}{2}} e^{\frac{9}{10}} \delta^{\frac{1}{R}} - (2e - 2)N^{\frac{1}{2}} - 1}{2e^2 N^{\frac{1}{2}}} \tag{53}$$

We then explicitly calculate the constants in the numerators, and slightly tighten the constraint by rounding the constants up/down appropriately. Since we are tightening the constraint, we can also replace the "less than" with a "less than or equal to." We finally get a simple constraint for $q$, such that any $q$ that satisfies the below inequality will solve the $\delta$ nearest neighbor problem:

$$q \leq \frac{4.80 N^{\frac{1}{2}} \delta^{\frac{1}{R}} - 3.44 N^{\frac{1}{2}} - 1}{2e^2 N^{\frac{1}{2}}} \tag{54}$$

Notice that this bound for $q$ is positive when $R$ is sufficiently large, since $\delta^{\frac{1}{R}}$ approaches 1 and the numerator approaches the positive value $1.36 N^{\frac{1}{2}} - 1$.

**Solving for $p$, $q$, and $R$:** We now fix $p$ and $q$ to be the largest (and thus, least expensive) values that obey their respective simplified constraints. Using the edge of the $1 - p$ constraint range from Equation (44) and the edge of the $q$ constraint range from Equation (54), we set

$$p = 1 - \frac{\delta}{2R} \tag{55}$$

$$q = \frac{4.80 N^{\frac{1}{2}} \delta^{\frac{1}{R}} - 3.44 N^{\frac{1}{2}} - 1}{2e^2 N^{\frac{1}{2}}} \tag{56}$$

Notice that – although our analysis is performed under the assumption that $R > 10 \log N$ – we may still choose a value for the free parameter $R$. Our strategy is to select a value of $q$ which satisfies the

---

[6]We make the substitution whenever the replaced term monotonically increases with increasing $N$ and $R$

constraint, and then solve for an $R$ which guarantees this value of $q$. To simplify our later analysis, we will use

$$q = \frac{1}{\sqrt{N}} \tag{57}$$

We can now plug in Equation (57) into Equation (56) and solve for $R$:

$$R = \frac{\log(\frac{1}{\delta})}{(\log(4.80N^{\frac{1}{2}}) - \log(2e^2 + 3.44N^{\frac{1}{2}}))} \tag{58}$$

Note the right side is a valid fraction less than 1 because $N >= 150$. In fact, the denominator is fixed within a small range: the largest it can be is when $N = 150$, when it is about 0.97, and the smallest it can be is about 0.72, when $N \to \infty$. Thus the denominator is $O(1)$.

When $\delta$ is small enough that $R > 10 \log N$, then Equation (58) yields a value of $R$ which satisfies our simplifying assumption and attains the correct value of $q$.

We have now found explicit values of $p$, $q$ and $R$, which constrain all of the free parameters of our data structure. These values are given by Equation (55), Equation (57), and Equation (58). We have shown that these values attain a sufficiently low false positive rate $Q$ and high true positive rate $P$ to solve the nearest neighbor problem, proving the theorem. □

We have specific parameter settings from Lemma 3 that solve the nearest neighbor problem, but it remains to bound the query time of the resulting data structure. We obtain our main theorem by using these values with the query time expresssion from Lemma 2.

**Theorem 4.** *(Main Theorem) Under the conditions in Lemma 3, we solve the randomized nearest neighbor problem for $\gamma$-stable queries in time $t_{query}$ with probability $1 - \delta$.*

$$t_{query} = O\left(N^{\frac{1}{2}+\gamma} \log^4(N) \log^3\left(\frac{1}{\delta}\right)\right) \tag{59}$$

*Proof.* We made many simplifications to the constraints for $p$ and $q$ in Lemma 3. These simplifications require us to solve a *harder* version of the problem than necessary. For example, there is some maximum value of $\delta$ in Lemma 3 that suffices to make $R \geq 10 \log N$. Call this $\delta'$, such that when $\delta = \delta'$ the value of $R$ from Lemma 3 is $10 \log N$. To solve the nearest neighbor problem for an arbitrary $\delta$, we split our analysis into two cases, $\delta \geq \delta'$ and $\delta < \delta'$, and solve for the query time under each one.

**Case 1, $\delta \geq \delta'$:** If $\delta \geq \delta'$, we use the values from Lemma 3 with $\delta = \delta'$. This will return an array of tests that solves the $\delta'$ nearest neighbor problem. Because we substantially simplified the constraints in Lemma 3, $\delta' < \delta$ and thus we solve a harder version of the problem than necessary. When $\delta = \delta'$, we have the following from Lemma 3:

$$R = 10 \log N$$
$$q = N^{-\frac{1}{2}}$$
$$1 - p = \frac{\delta'}{2R}$$
$$|K| = 1$$

We now plug these values into our query time result from Lemma 2 and simplify. Note that Lemma 2 has its *own* failure probability $\delta_3$. Note that to have $R = 10 \log N$, $\frac{1}{\delta'}$ is slightly smaller than $N^{10}$, so $\log(\frac{1}{\delta'}) = O(\log N)$. This leaves us with

$$t_{query} = O(N^{\frac{1}{2}+\gamma} \log^4(N) \log(1/\delta_3)) \tag{60}$$

**Case 2, $\delta < \delta'$:** If $\delta < \delta'$, then our simplifying changes to the constraints no longer force us to solve a harder problem than necessary. In this case, we use the values from Lemma 3 using $\delta$. As before, we obtain our query time result from Lemma 2:

$$t_{query} = O\left( \log\left(\frac{1}{\delta}\right) N^{\frac{1}{2}+\gamma} \log^2(N) \right.$$
$$\left. \max\left(\log(N^{\frac{1}{2}}), \log\left(\frac{R}{\delta}\right)\right) \log(1/\delta_3) \right) \tag{61}$$

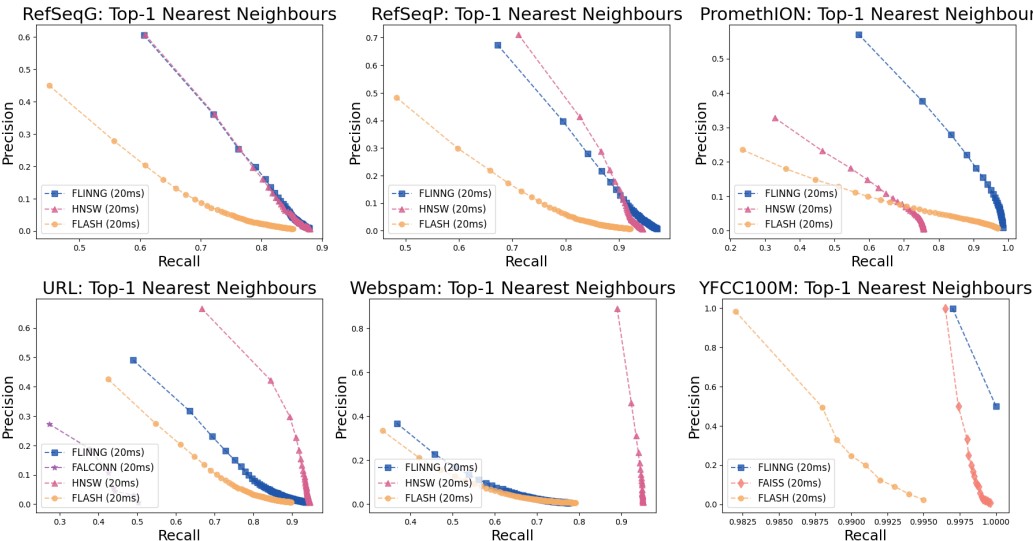

Figure 1: Precision recall graphs for the top 1 nearest neighbours on each dataset.

Table 1: Algorithm index sizes in gigabytes, determined by the minimum index size an algorithm achieved over all hyperparameters, subject to having recall greater than $R$. All recalls are R1@100 except for YFCC100M, which is R1@1.

| DATASET | $R$ | FLINNG | FAISS | FALCONN | HNSW | FLASH | INV. | GROUPS |
|---------|-----|--------|-------|---------|------|-------|------|--------|
| REFSEQG | 0.8 | 0.090 | - | - | 0.031 | 4.29 | - | - |
| REFSEQP | 0.8 | 0.090 | - | - | 0.031 | 4.29 | - | - |
| PROMETHION | 0.8 | 0.074 | - | - | 0.64 | 8.59 | 4.22 | - |
| URL | 0.5 | 0.048 | - | - | 3.03 | 4.29 | 2.21 | - |
| WEBSPAM | 0.5 | 0.0068 | - | 2.25 | 8.04 | 4.29 | - | 0.14 |
| YFCC100M | 0.95 | 3.5 | 3.7 | - | - | 4.29 | - | - |

Note we used the fact that $R = O(\log(\frac{1}{\delta}))$, since as we noted in the proof of Lemma 3 the denominator in the equation for $R$ is $O(1)$. We can further simplify Equation (61) by rewriting $\log\left(\frac{R}{\delta}\right)$ as $\log(R) + \log\left(\frac{1}{\delta}\right)$, and recognizing that $\log(R)$ is dominated by $\log\left(\frac{1}{\delta}\right)$. Furthermore, since both terms in the $\max$ operation are greater than 1, we note that the maximum is asymptotically smaller than the product of the two terms. After simplifying, we have that

$$E[t_{query}] = O\left(\log^2\left(\frac{1}{\delta}\right) N^{\frac{1}{2}+\gamma} \log^2(N)\right) \tag{62}$$

**Combining Results:** In total the runtime for an arbitrary $\delta$ is the maximum of case 1 (Equation (60)) and case 2 (Equation (62)):

$$t_{query} = O\left(N^{\frac{1}{2}+\gamma} \log^4(N) \log^2\left(\frac{1}{\delta}\right)\right) \tag{63}$$

Here, we have absorbed $\delta_3$ into $\delta$, which adds only a constant multiplier to the expression. $\qquad\square$

## 2 Experiments

In this section, we provide additional details about our experiments. We also show a full table of index characteristics for each dataset in our evaluation.

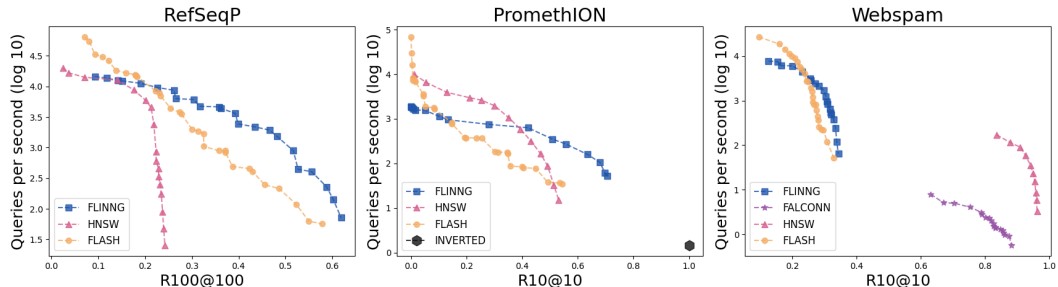

Figure 2: Plots for top-10 and top-100 nearest neighbor search on selected datasets. FLINNG performs best when the top neighbors are very similar to the query, as predicted by the theory.

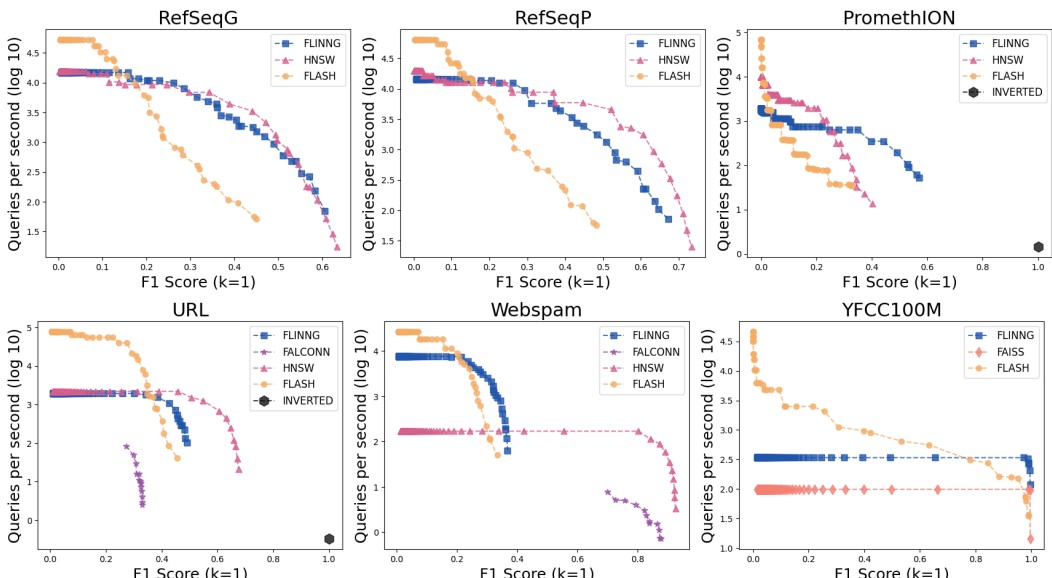

Figure 3: F1 scores for top-1 nearest neighbor search on all datasets.

## 2.1 Dataset Details

We used the following metric spaces for the datasets in our experiments. For RefSeqG, RefSeqP and PromethION, we use the Jaccard similarity space. For the purposes of Jaccard similarity, the tokens associated with a genetic sequence are the sets of all $k$-mers (size $k$ sub-strings) present in the sequence. We use MinHash as the hash function. For distance-based indices such as HNSW and FALCONN, we preprocess the data by computing 100 MinHash signatures and estimate the Jaccard similarity rather than compute it explicitly, as this is more efficient.

For the Webspam and URL datasets, we consider the cosine similarity between the high-dimensional and sparse representations of the data. Here, we use MinHash as the hash function. For YFCC, we consider the Euclidean distance and use signed random projections.

## 2.2 System Details

We performed all experiments using $1.48$ TB of RAM. For YFCC100M, we used 88 Intel Xeon E5-2699A v4 processors, each of which has a $56$ MB L3 cache. For YFCC100M, we used 96 Intel Xeon Gold 5220R processors with a 36 MB cache.

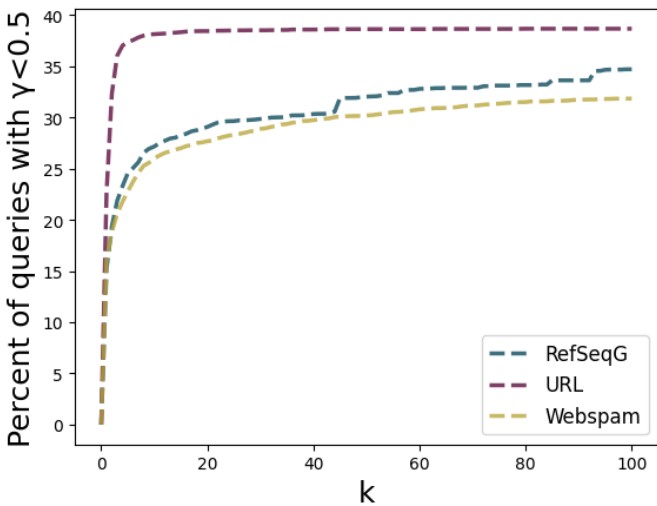

Figure 4: Percentage of benchmark queries with $\gamma < 0.5$, for some cutoff number of neighbors $K$ less than or equal to $k$.

## 2.3 Baseline Failures

Some of the baseline methods did not run on some of the datasets. We attempted to construct HNSW and FALCONN indices on YFCC100M, but memory limitations meant that the index would not fit within our 1.48 TB of RAM. We modified the HNSW library to work with nonstandard short floating point vectors, but the resulting index took more than 5 days to construct. We also tried to build FAISS indices on the genomics and web datasets, but were unable to fit the quantized data in memory because these problems are ultra high-dimensional.

## 2.4 Hyperparameters

For algorithms with fast indexing times like FLINNG, FLASH, FALCONN, and the grouping algorithm from [4], we tried hundreds of hyperparameter settings and selected the best indices. For algorithms such as HNSW and FAISS, which can take hours or days to train, we built indices using suggested parameters and tuned the query-time arguments aggressively.

FLINNG requires four hyperparameters: $R$, $B$, $m$, and $L$. We use $R = \{2, 3, 4\}$ and $B = 2^a$ for $a \in [11, 15]$. To have 16-bit cell IDs, we constrain $BR < 2^{16}$. For YFCC100M, we use $R = 2$ and $a \in [12, 19]$. We set the number of LSH functions $m$ to $2^a$ for $a \in [2, 11]$. We used $L = 18$ for Webspam, $L = 12$ for YFCC100M and $L = 17$ for the other datasets.

FLASH requires the following hyperparameters: $m$ (the number of hash tables), $L$ (the number of hash functions in each table), and $s$ (the size of each reservoir for reservoir sampling). We used the recommendations from the paper. However, we found that much larger values of $m$ and $s$ were needed than in the original paper to obtain high recall on some of our tasks. We used the same values of $m$ as the authors of FLASH: $m = 2^a$ for $a \in [2, 11]$ for all datasets. For the non YFCC100M datasets, we let $s = 2^a$ for $a \in [2, 11]$. For YFCC100M, we let $s = 2^a$ for $a \in [3, 12]$. As with FLINNG, we used 18 hash bits for webspam, 12 hash bits for YFCC100M, and 17 hash bits for every other dataset.

Our implementation of the grouping algorithm from [4], which we denote GROUPS, requires $M$, the number of groups, $t$, the number of back propagation steps, $N_L$, the number of groups each point is in, and $R$, the total number of points to checks across all $t$ back propagation steps (see the original paper for more details on each parameter). None of our datasets in dense format fit in memory, which GROUPS requires, but we were able to project the URL and Webspam datasets into 400 dimensions using 400 random projections to get a meaningful benchmark against our algorithm (we cannot apply random projections to the other datasets so we were not able to run GROUPS on the other

datasets). For Webspam, we tried all combinations of $M = 20000, 40000$, $t = 1, 2, 4$, $N_L = 2, 4$, and $R/t = 10000, 80000$. For URL, we tried all combinations of $M = 20000, 80000, 320000$, $t = 1, 2, 4$, $N_L = 2, 4$, and $R/t = 10000, 20000, 80000$.

FALCONN could only run on the URL and Webspam datasets because the package does not natively support Jaccard similarity for the genome datasets and has out-of-memory issues for YFCC100M. FALCONN requires three hyperparameters: $m$ (the number of hash tables), $n_p$ (the number of probes for multi-probe LSH), and $L$ (the number of hash functions for each table). We let $n_p = 2^a$ for $a \in [1, 9]$, $m = 2^a$ for $a \in [1, 4]$, and used $22, 20, 18$ hash bits for URL and $20, 18, 16$ hash bits for Webspam.

Due to the high dimensionality of the other datasets, FAISS was only feasible for YFCC100M. For high-dimensional sparse data such as Webspam or URL, quantization actually *increases* the memory of the index. We used an inverted file index with product quantization. This index requires two construction parameters: $m$ (the number of $k$-means centroids) and $s$ (the number of bits for product quantization). There is one query-time parameter $n_p$ (the number of clusters probed for each query). FAISS also supports the use of an HNSW graph to identify the best clusters, so we use both flat (i.e. brute force) and HNSW pre-indexing. We trained 3 different indices, all with $s = 32$ bit product quantization: $m = 4$k centroids with flat (brute force) indexing, $m = 262$k centroids with flat indexing, and 65k with HNSW indexing. We used a subset of one million points to train the $k$-means centroids. We used $n_p = 2^a$ for $a \in [1, 9]$. We observe that all of these IMI indices performed about the same. We calculated the Pareto frontiers by varying $n_p$ (the primary way to trade recall for search time in FAISS) and reported the best results in our experiments. We would also like to note that our query comparisons were done on CPU, one query at a time, as is standard when benchmarking near neighbor algorithms [1]. However, for fairness we exclude the $k$-means training time from the index construction time as this can be efficiently done on GPU.

HNSW requires two construction hyperparameters: $ef_c$ and $M$. The parameter $M$ is the maximum number of edges for each node in the graph, while $ef_c$ may be thought of as a parameter that controls the quality of the near neighbor graph (larger is better). We used parameter $M = 32$ and $ef_c = 100$ for all trials. HNSW has one query-time parameter $ef_s$, which controls the recall-time tradeoff. We let $ef_s = 2^a$ for $a \in [4, 14]$. For our genomics datasets (PromethION and RefSeq), we used a pregenerated and fixed number of minhashes to allow HNSW to perform fast search in the Jaccard metric space. We used $2^a$ for $a \in [1, 11]$ number of hashes. We modified the HNSW code to support the approximate Jaccard metric by implementing a distance functin that counts the collisions among these hashes for two sequences. We tried to build an index for YFCC100M, but the graph construction algorithm did not finish,even after four days of construction time.

Finally, we used inverted indices to compute the ground truth results for the URL and PromethION datasets. We show the query time for this structure as a baseline. The other genomic datasets (RefSeqG and RefSeqP) were too high-dimensional for an inverted index lookup to be practical, and Webspam and YFCC100M had too many nonzeros.

## 2.5  Supplementary Plots

Figure 1 shows the precision recall plots for all datasets considered in our evaluation. Figure 2 shows the latency-recall relationship for top-10 and top-100 near neighbor search on selected datasets. Figure 3 shows the F1 scores for each index, and Figure 4 shows the empirical distribution of $\gamma$ values for three datasets. Note that a surprisingly large fraction of queries satisfy our theoretical assumptions - up to 40%.

## 2.6  Index Characteristics

Table 1 shows the index size for all of the indices considered in our evaluation.