# OpenReview forum: " Practical Near Neighbor Search via Group Testing"
_NeurIPS.cc/2021/Conference — NeurIPS 2021 Spotlight_

### Official Review · Reviewer_eTUm · 2021-07-09

**Rating:** 7
**Confidence:** 4

**Summary:**

The paper develops a new algorithm for approximate near neighbor search by combining ideas from group testing and bloom filters. The proposed algorithm called FLINNG casts the problem of finding the near neighbor of a query point in a high-dimensional dataset, as the problem of finding a few number of "positive" groups in a group testing problem where each group captures the absence or presence of at least one near neighbor of the query in a small random subset of data points. The paper provides theoretical justifications of the sublinear query time of the algorithm in number of data points and provides empirical evidence on real-world data sets which demonstrates comparable to superior near neighbor query performance compared to some to some baseline algorithms.

**Ethical Concerns:**

Non.

**Limitations And Societal Impact:**

Yes.

**Main Review:**

The paper provides a novel method for the approximate near neighbor problem. The main advantages is that the query time (under some data-dependent assumptions) is sublunar (and therefore fast) and also the implementation is easy and in particular alleviates the need for clustering scheme or graph algorithms. However, the simplicity is highly conditioned on the data-dependent assumption. Therefore, the main limitation is on the gamma-stability requirement of the algorithms which states that the query point should be much more similar to a small subset of size |K| of the data points and be sufficiently far from the rest. This is needed to do the analysis since such data-dependent criteria ensures some notion of sparsity both in the group testing and Bloom filtering stage. My detailed comments are as follows:

Theoretical comparisons. A comparison table containing the theoretical bounds in terms of (at least) the query time and (ideally) the memory and indexing time can be shown to compare FLINNG with other baseline algorithms (including the LSH). This will strengthen the paper and clarify the theoretical contributions. Is there any other sublinear time near neighbor algorithm that employs the stability condition in its bound?

Utility of gamma-stable condition. To ensure the sublinearity claim gamma needs to be smaller than ½ in equation (7); enforcing this condition in Definition 3 yields the condition that s_{|K|+1} < 1/s_{|K|}. For large values of s_{|K|} this will results in very small values for s_{|K|+1}. To convince the utility of the gamma-stable condition it would be helpful to plot the empirical distribution of gamma (if obtainable) for practical values of |K| (e.g., |K| \in {5,10,50,100}) in a few (maybe small) datasets. This is important since the sublinearity bound highly depends on that. Data that are highly clustered clearly pass the condition; an empirical analysis of the datasets listed in the paper can validate that for the real-world datasets in the paper. Also is there any faster way to check if the stability condition is satisfied for a dataset?

Disparity across datasets.  It would be helpful if the authors can explain conditions under which HNSW can significantly outperform FLINNG, see e.g., Figure 5 in the appendix (Webspam Top-1 NN and RL10@10).

Lemma 1. In lemma 1 does increasing R decrease the lower bound for true positive rate?

Index characteristic. It would be important to add the Index characteristic of Table 2 for other data sets to the appendix.

F-score. Since both precision and recall are critical in these tasks is it possible to add the time against F1 score in Figure 3?

Small typos: line 271: “to achieve good practical”


**Time Spent Reviewing:**

5

---

> ### Author Response · Authors · 2021-08-10
> **Thank you for pointing out places to improve the work**
>
> Thank you for your review - we address your points (in order) below.
>
> ### 1. Stability and Theoretical Guarantees
> The notion of stability explicitly comes into the picture for most LSH-based algorithms and is also present for other strategies (though not always explicitly, or in the same form as for our method). Here is a summary, which we'll put into a table in the revision:
>
> (A) Adapted to our problem setting, conventional bucket-based LSH algorithms have complexity $N^{\rho}$, where $\rho =  \frac{\log 1/s_{|K|}}{ \log 1/s_{|K|+1}}$. This comes from the traditional LSH analysis, which shows that we need $L = N^{\rho}$ hash functions. The analysis proceeds by bounding the collision probabilities $s_{|K|}$ and $s_{|K|+1}$ to bound $\rho$ for specific hash functions and metric spaces [1,2].  A large amount of work has gone into finding algorithms with provably large $s_{|K|}$ and small $s_{|K|+1}$ that match the optimal bounds - this work also benefits our method since we too need large $s_{|K|}$ and small $s_{|K|+1}$ [3,4,5].
>
> (B) While not a sub-linear time algorithm, the space complexity of [6] (a non-traditional LSH-based algorithm) is $N^{\rho}$ where $\rho = \frac{(6 \log s_{|K|} + o(1) )} {\log s_{|K|} - \log s_{|K|+1}}$. The time complexity is $N^{1+\rho}$ but heuristics make it reasonably fast in practice.
>
> (C) For random partition trees and KD-trees, the failure probability is proportional to $\Phi$, where $\Phi$ is a stability parameter that is small when the nearest neighbor is significantly closer than the rest of the data and 1 when all points are the same distance from the query. If the failure probability is held constant, then the query time depends on $\Phi$ [7]. Specifically, if $x^{\star}$ is the neighbor then
> $$ \Phi(q) = \frac{1}{N-1} \sum_{x_i \neq x^{\star}}\frac{\mathrm{dist}^2(q,x_i)}{\mathrm{dist}^2(q,x^{\star})} $$
>
> (D) There are no general results for graph-based algorithms, but recent work on uniformly-distributed spherical data shows a hardness gap between “dense” and “sparse” regimes. While the guarantees cannot be directly compared, this corresponds to our stability criterion in the sense that (with high probability) neighbors in the sparse regime are stable and neighbors in the dense regime are not stable [8].
>
> ### 2. Definition of Gamma and Practical Values of Gamma
> The definition of $\gamma$ is missing a negative sign - we apologize for the typo. After the typo correction, your condition becomes $s_{|K|}^{1 + \gamma} \leq s_{|K|+1}^{\gamma}$, which is much more reasonable. For example if $s_{|K|} = 0.9$, then we need $s_{|K|+1} \leq 0.729$ to have $\gamma \leq 0.5$.
>
> We agree that empirical values of $\gamma$ (for different K) are interesting. OpenReview will not allow us to attach a plot, but here are some empirical values. The first number is the percentage of our 10k queries that have $\gamma < 0.5$ for K = 1. The second number is the percentage that have $\gamma < 0.5$ for some K between 1 and 100:
>
> Genomes: (15%, 35%)
> Proteomes: (10%, 20%)
> Webspam: (16%, 30%)
> Url: (22%, 38%)
>
> Thus, it is reasonable to expect (at least some) practical queries to satisfy the condition. Of course, in practice FLINNG works very well even on queries that do not meet the gamma-stable condition - this is because the theory makes several pessimistic assumptions (e.g. that all non-neighbors have similarity $s_{|K|+1}$). This is necessary for worst-case guarantees but does not happen in practice.
>
> One option to quickly check whether the data satisfies the stability condition is to compute ground-truth neighbors for a few queries and explicitly compute gamma, but this is not very satisfying. However, it might be possible to use an LSH table of the dataset, by finding the mean and population of each non-empty bucket and computing approximate distances between clusters. This is an interesting direction which we plan to test.
>
> ### 3. Disparity across Datasets
> This could be due to FLINNG's use of MinHash as a proxy for angular distance, leading to a mismatch between distance and hash function. We did this because MinHash is reported to be very fast for sparse data (see [9]), but we plan to rerun the algorithm with (sparse) signed random projections to see whether that changes the performance on webspam and url.
>
>
> ### 4. Comment on Lemma 1
> Yes, increasing R decreases the lower bound. This makes intuitive sense, since increasing the number of rows increases the number of group tests that must be positive for a point to be a positive. It also decreases the upper bound for the false positive rate for the same reason. Our theoretical analysis shows how to balance this tradeoff.
>
> ### 5. Index sizes
> Table 3 lists index sizes for all algorithms on all datasets. We are happy to rerun experiments to get the build times, if necessary, but we believe they will follow a similar pattern as YFCC.
>
> ### 6. F1 scores
> Yes, we will add these graphs in the revision. We have already created the graphs but cannot attach them here, so instead we report some specific values for F1 scores at time thresholds:
>
> Promethion@20 ms, K = 1: FLINNG F1 = 0.6, next best (HNSW) F1 = 0.4
>
> Promethion@20 ms, K = 10: FLINNG F1 = 0.7, next best (HNSW) F1 = 0.55
>
> YFCC, K = 1: FLINNG F1 ~= 1 at ~300 queries per second, next best (FAISS) F1 ~= 1at ~100 queries per second
>
> Webspam, K = 1: For 2.2 < log_10(queries) < 3.8, FLINNG has the best F1 score (around 0.38) by about 0.05 (the next best is FLASH with around 0.33).
>
> Webspam, K = 1: For log_10(queries) < 2.2, there's a sharp change in behavior: HNSW can very quickly achieve an F1 score > 0.9.
>
> ## 7. Typos
> Thank you for pointing out this typo, we will fix it in the final camera ready version.
>
>
> ## References
>
> [1] Indyk, Piotr, and Rajeev Motwani. "Approximate nearest neighbors: towards removing the curse of dimensionality." *Proceedings of the thirtieth annual ACM symposium on Theory of computing. 1998.*
>
> [2] Gionis, Aristides, Piotr Indyk, and Rajeev Motwani. "Similarity search in high dimensions via hashing." *Vldb. Vol. 99. No. 6. 1999.*
>
> [3] Andoni, Alexandr, and Piotr Indyk. "Near-optimal hashing algorithms for approximate nearest neighbor in high dimensions." *2006 47th annual IEEE symposium on foundations of computer science (FOCS'06). IEEE, 2006.*
>
> [4] O’Donnell, Ryan, Yi Wu, and Yuan Zhou. "Optimal lower bounds for locality-sensitive hashing (except when q is tiny)." *ACM Transactions on Computation Theory (TOCT) 6.1 (2014): 1-13.*
>
> [5] Andoni, Alexandr, and Ilya Razenshteyn. "Optimal data-dependent hashing for approximate near neighbors." *Proceedings of the forty-seventh annual ACM symposium on Theory of computing. 2015.*
>
> [6] Coleman, Benjamin, Richard Baraniuk, and Anshumali Shrivastava. "Sub-linear memory sketches for near neighbor search on streaming data." *International Conference on Machine Learning. PMLR, 2020.*
>
> [7] Ram, Parikshit, and Kaushik Sinha. "Revisiting kd-tree for nearest neighbor search." *Proceedings of the 25th acm sigkdd international conference on knowledge discovery & data mining. 2019.*
>
> [8] Prokhorenkova, Liudmila, and Aleksandr Shekhovtsov. "Graph-based nearest neighbor search: From practice to theory." *International Conference on Machine Learning. PMLR, 2020.*
>
> [9] Anshumali Shrivastava and Ping Li. 2014."In defense of minhash over simhash." *In Artificial Intelligence and Statistics. 886–894*

---

> > ### Comment · Reviewer_eTUm · 2021-08-18
> > **reply**
> >
> > Thanks for the reply. Please make sure 1) the appropriate corrections are made in the manuscript, 2) the suggested plots are added, and 3) some discussion is made towards other theoretical works. I am willing to increase my score in light of the rebuttal.

---

### Official Review · Reviewer_sLDs · 2021-07-16

**Rating:** 6
**Confidence:** 4

**Summary:**

The paper tackles the problem of near-neighbor (and nearest-neighbor) search in high-dimensional spaces. The authors propose to use group-testing on top of distance-sensitive Bloom filters: to simplify, the returned set is the intersection of the union of small sets, each of these small sets being a hash bucket that the query falls into. The authors derive the theoretical properties of their method FLINNG (sublinear query time), and show the superiority of their method compared to the state of the art (FAISS, HNSW, FLINNG) through extensive experiments on genome, URL and embedding data.

**Limitations And Societal Impact:**

Yes

**Main Review:**

The paper is interesting and well-written, and tackles an important problem.
The construction of the algorithm is smooth to follow at a high-level, even though the meat resides in being able to construct a classifier with given true and false positive rates: to do this in a reasonable time, the authors have to assume gamma-stability which is a strong hypothesis. I was very positively surprised by the very strong empirical performance of the algorithm. I wonder if it is biased compared to other methods that operate on embeddings, because the datasets considered are mostly long sequences which means very high-dimensional data.
I have a couple of questions for the authors:
- Did you try other embedding datasets that YFCC100M, like Deep1B or BigANN1B?
- How do you go from near- to nearest-neighbor search? In the paper, you mention running Algorithm 3 with various thresholds but Algorithm 3 doesn’t seem to take a threshold as input.
- What setting of FAISS did you use? FAISS is a library implementing multiple quantization-based algorithms, I am assuming this is IMI (Inverted Multi Index)?


**Time Spent Reviewing:**

2

---

> ### Author Response · Authors · 2021-08-10
> **Answers to questions and clarification about threshold relaxation**
>
> Thank you for the helpful and thoughtful comments. We answer your points in order below.
>
> 1. We expect our algorithm to perform better on sparse, high dimensional data (> 500 dimensions or so). We expect HNSW and other graph based indices to perform better on medium-dimensional embeddings like Deep1B and BigANN1B (this will almost certainly be the case, though we're happy to run the experiment. In general, LSH-based methods are typically only good for high dimensional data). We developed FLINNG because we feel that the high-dimensional case is sufficiently important to merit individual consideration. See [1] and [2] for applications where high (> 500) dimensional embeddings lead to improved performance. These systems are bottlenecked precisely by the failure of traditional methods on super high-dimensional inputs.
>
> [1] Nigam, Priyanka, et al. "Semantic product search." KDD 2019.
> [2] Medini, Tharun, Beidi Chen, and Anshumali Shrivastava. "SOLAR: Sparse Orthogonal Learned and Random Embeddings." ICLR 2020.
>
> 2. Algorithm 3 is not run with different thresholds; instead it progressively relaxes an (implicit) internal threshold until it returns the desired number of points. Algorithm 3 works by selecting cells (i.e. marking them as “positive”) in descending order of Bloom filter collisions. This effectively relaxes an internal threshold because we start by selecting only cells with 100% collisions, later selecting ones with a lower number of collisions. We stop once the union of cells contains enough points.
>
> 3. Yes, this is using an IMI. As mentioned in the supplementary materials, we used an inverted file index with product quantization. We tried 3 different FAISS indices, all with s = 32 bit product quantization: m = 4k centroids with flat (brute force) indexing, m = 262k centroids with flat indexing, and 65k centroids with HNSW indexing. All had similar performance. See the supplementary materials, specifically the paragraph starting at line 759, for further details.

---

> > ### Comment · Reviewer_sLDs · 2021-08-30
> > **Thanks to the authors for the response.**
> >
> > Thanks for the response.
> > Given the response as well as other reviews, I maintain my original rating. I think the paper is valuable and makes good contributions, in particular I appreciate that the authors provided experimental values for gamma: assuming Definition 3 seemed to be a strong requirement but seems to happen in practice more than what I would have expected. I would say that one limitation is still the small experimental advantage in some setups (depending on the datasets/recall levels, it can be close to or below HNSW or FLASH) but that does not prevent the paper for being overall strong.

---

### Official Review · Reviewer_e9ca · 2021-07-16

**Rating:** 6
**Confidence:** 4

**Summary:**

Near Neighbor Search (NNS) is one of the most fundamental problems in machine learning. This paper proposes an efficient NNS method for high-dimensional data by transforming the original problem into a group testing problem. The authors provide a detailed theoretical analysis of the distance-sensitive Bloom Filters for group testing to demonstrate how the query time and the quality are bounded with proper parameters settings. Experiments validate the performance of FLINNG.

**Ethical Concerns:**

No.

**Limitations And Societal Impact:**

Overall, I like this paper. However, I still have some questions about this paper, and some limitations justify my current rating.

1. The $\gamma$-stable query condition is a relatively strong requirement for the query. As pointed out by the authors, FLINNG works better for genomes data such as RefSeqG and RefSeqP that have ultra-high dimensions. However, for other ultra-high dimensional datasets such as Webspam and URL, the performance of FLASH is better than that of FLINNG, which might be due to this strong query condition. This limit this work to be applied to other applications. Or could you further analyse whether there is any mathematical condition to indicate a dataset as suitable for this method? You can assume the query and data are from the same distribution.

2. For the $m$ hash tables used for the distance-sensitive Bloom filter, the bound of the hash code $h(x)$ is not clearly explained. Since the FLINNG structure is an inverted index (as indicated in Section 6) from hash code to cell, the range of $h(x)$ would greatly affect the space requirement of the index structure. This may need to explain more clearly.

3. The hash functions and distance metrics used for each dataset are not specified. For example, the authors only mentioned some of the datasets are compressed via MinHash, but they did not specify the LSH for the distance-sensitive Bloom filter. I suppose MinHash and Jaccard similarity is applied for the sparse datasets, but what about the dense dataset YFCC100M? And FAISS do not support Jaccard similarity.

4. In the experiments, some necessary information is missing.
(1) The index construction time comparison with FAISS lack details such as whether the GPU support of FAISS is enabled. Also, since there are plenty of parameter combinations, what exactly is the trade-off between search performance and index constructions.
(2) The number of queries is not specified for each dataset. For query time, since FAISS can support batch queries so whether the comparison is under maximum query batch size or just one query at a time would greatly affect the query efficiency.

5. The paper presentation can be further improved. For example, In Section 5, some of the references are clearly wrong and get mixed up with the proof in Appendix, e.g., In Lemma 2 (Line 258), they claim "Under the assumptions in Lemma 2..." Additionally, there is a reference to Figure 4 in the Appendix but Figure 4 is missing.


**Main Review:**

1. I appreciate the idea of using group testing to shortlist the nearest neighbour candidate and use repetition to remove false positives.

2. FLINNG enjoys a provable bound for query time and quality.

3. This work provides a detailed theoretical analysis to guide parameter settings in the experiments.

4. The paper is easy to follow, and Figure 1 is clear to show the general workflow of the proposed method.


**Time Spent Reviewing:**

10

---

> ### Author Response · Authors · 2021-08-10
> **Answers to questions about stability, experiments, and hash functions**
>
> Thank you for your detailed comments and the large amount of time you spent considering our work. Please find the answers to your points below.
>
> 1. One favorable data / query distribution is high-dimensional data with tight clusters. To see this, note that our gamma-stable condition can also be written as $s_{|K|}^{1 + \gamma} \leq s_{|K|+1}^\gamma$ (For details, see response to reviewer eTUm). This condition means that FLINNG is good for any distribution that emits small clusters (of size |K|) which are mutually far from other clusters. The clusters must be tight enough that the similarity within the cluster is greater than $s_{|K|}$ and well-spaced enough that other clusters have lower than $s_{|K|+1}$ similarity. Several interesting distributions can satisfy this assumption. For example, a Gaussian mixture model with sufficiently well-spaced means and small variance will have gamma-stable queries. It should be possible to bound the variance when the Gaussians are placed on a lattice or to tie our cluster-based intuition to a measure of cluster quality (one of the many clustering objectives, perhaps), though this isn’t completely straightforward.
>
> 2. We use the typical LSH analysis, which is not affected significantly by the range of $h(x)$ because the analysis interacts with the hash function only through its collision probability. In practice, LSH functions can have a large range, but practitioners frequently take the modulo or rehash into a small range. It is known that such rehashing only adds small, linear, constant distortions to the collision probability. This is easy to include and does not change our conclusions (the new total space complexity is $O(NmBr) = O(N^{\frac{3}{2}}log^2(N)))$. We are happy to address this in the paper if needed.
>
> 3. Thank you for pointing this out, it seems we mistakenly cut some lines in the appendix. The hash function families and distance metrics by which we evaluate ground truth are as follows:
>
> RefSeqG, RefSeqP, Promethion: Minhash, Jaccard similarity
> Webspam, URL: Minhash, cosine similarity
> YFCC: Signed Random Projections, Euclidean distance
>
> Jaccard similarity only makes sense for sparse data, and FAISS cannot run on most sparse datasets because it internally uses a dense format that runs out of memory. Thus, we did not run into the problem that FAISS does not support Jaccard similarity.
>
> 4. During index construction we did not enable GPU support for FAISS (this mainly affects training time). For fairness, we exclude this training time from our FAISS comparison. As mentioned in supplementary material, we did try 3 different indices, with s = 32 bit product quantization: m = 4k centroids with flat (brute force) indexing, m = 262k centroids with flat indexing, and 65k centroids with HNSW indexing.  We observe that all three of these IMI indices performed about the same (the pareto frontier is from n_p, the number of clusters probed, which is the primary way to trade recall for search time). Thus, we do not believe that parameter sensitivity could be a major issue.
>
> We tested with 10,000 queries for every dataset. Following standard practice, see e.g. ANN-Benchmarks [1]), we tested one query at a time on CPU for all the methods. It should be noted that all near-neighbor query algorithms are embarrassingly parallel, so batching can be leveraged for all baselines. This is the reason for our focus on single query latency and is likely the reason why standard benchmarks do the same.
>
> [1] Aumüller, Martin, Erik Bernhardsson, and Alexander Faithfull. "ANN-benchmarks: A benchmarking tool for approximate nearest neighbor algorithms." In International Conference on Similarity Search and Applications, pp. 34-49. Springer, Cham, 2017.
>
> 5. Thank you for finding this typo - we will fix this in the final version. To clarify: Theorem 2, Lemma 2, Lemma 3, and Theorem 4 (lines 251, 258, 264, and 268) should say “Under the assumptions in Lemma 1”. Line 266 should read “We obtain our main theorem by using the values from Lemma 3 with the query time from Lemma 2.” Regarding the appendix, the labels “Figure 4” and “Figure 5” both refer to Figure 5 (which we will re-label and fix).

---

> > ### Comment · Reviewer_e9ca · 2021-08-24
> > **Thank you for the detailed answers**
> >
> > Thank you for the detailed answers to my concerns.
> >
> > Questions 1 and 2 are clear to me. I suggest the authors include these explanations to the paper.  For Questions 3-5, thank you for your clarification and explanation, and I tend to believe the paper presentation (especially the theoretical part) can be improved. One minor issue is that the signed random projection LSH family might be not suitable for Euclidean distance. I suggest the authors try [1] or its variants for YFCC100M.
> >
> > Overall, the feedback makes this work more convincing to me. I suppose the authors could improve the paper and add more discussion in the final version by incorporating the comments from all reviewers, so I am glad to increase my rating. Thanks.
> >
> > [1] Datar, Mayur, Nicole Immorlica, Piotr Indyk, and Vahab S. Mirrokni. "Locality-sensitive hashing scheme based on p-stable distributions." In Proceedings of the twentieth annual symposium on Computational geometry, pp. 253-262. 2004.

---

### Official Review · Reviewer_557Q · 2021-07-16

**Rating:** 8
**Confidence:** 4

**Summary:**

In this paper, the authors presented an algorithm coined Flinters to Identify Near-Neighbor Groups (FLINNG) to solve the approximate near neighbor search problem. Given a dataset $D$, FLINNG is capable of constructing an {\it index} which, when given a {\it query} $y$, outputs a set of points $x_i \in D$ with high similarity to $y$.

The near neighbor search is a problem with extensive research and a wide range of applications, including recommendation systems, social networks, computer vision, and genome sequencing. While the problem is particularly well-understood in low dimensions, with efficient algorithms to identify exact k-neighbors, but large-scale applications on big datas with high dimentionality often challenges existing algorithms and data structures.

The authors address this need by showing that FLINNG is an index with efficient construction time, query time, simple structure, and low memory requirement. FLINNG utilizes many previously developed techniques, most notably a reduction from approximate near neighbor search to approximate set membership, and solving the latter using distance-sensitive Bloom filters, with a lower bound on the true positive rate and a upper bound on the false positive rate.

To construct the index, FLINNG randomly distributes all data points in $D$ evenly among $B$ groups, and repeats $R$ times to create $R$ such independent instances. For each of the $B \cdot R$ groups created this way, FLINNG constructs a classifier using distance-sensitive Bloom Filters with appropriate error guarantees, which solves the approximate set membership problem on this group.

When a query on point $y$ arrives, FLINNG queries each of the $B \cdot R$ classifiers on whether the corresponding group contains a near neighbor of $y$. If the classifier of a group returns an affirmative result, that group is likely to contain a near neighbor of $y$, and is considered a {\it candidate group}. FLINNG then first takes the union of all candidate groups within the same instance, then eliminates noises and amplifies accuracy exponentially by taking the intersection of the unions across all $R$ instances. The resulting intersection is returned as the near neighbor set of $y$.

The authors show that FLINNG can be highly efficient due to the properties of distance-sensitive bloom filters, which allows queries to have constant memory allocation and sublinear time complexity. An implementation is designed and compared with other status-quo near neighbor search algorithms like FLASH, HNSW, and FAISS, on the PromethION data set. These experiments show that indices constructed with FLINNG exhibit a 2x to 10x speed up in queries, as well as decreased index size and construction time.


**Limitations And Societal Impact:**

Yes

**Main Review:**


Overall, I believe that the article is well-written, well-thought, and of significance. The authors explained in detail the background, prior work, and significance of the problem and their solution, and provided theoretical and empirical evidence of the effectiveness. Although the main body of the article only stated the important theorems and lemmas vital to the analysis, a full proof is indeed provided in the appendix, complementing and completing the authors' argument.

2.1 Originality

The authors did a great job outlining the prior works done on the near neighbor search problem, with adequate citations. FLINNG utilizes the idea of group testing, which differs from a lot of status quo algorithms based on locality sensitive hashing, sample compression, or graph-based methods. Although there are prior works which apply group testing to near neighbor search, these methods are limited in feasiblity and effectiveness. The authors utilized distance-sensitive Bloom filters to circumvent the aforementioned limitations, resulting in an algorithm that performs better than its predecessors.

2.2 Quality

The work is technically sound to my best knowledge. The authors claimed that FLINNG is an algorithm that solves the approximate near neighbor search problem efficiently and effectively, and is comparable and competitive against all the status quo algorithms on near neighbor search. The authors then indeed supported these claims by theoretical analysis as well as empirical data. Theoretically, the overall structure of their analysis is sound, and their results are supportive of their claim: FLINNG can support queries with sublinear time complexity, and the error rate is exponentially small. Following these analysis, the authors provided experimental results, comparing the runtime and accuracy of FLINNG to other status quo algorithms including FLASH, FAISS, and HNSW. Their results show that FLINNG obtains a significant speed up and a decrease in data structure size compared to other algorithms. To my best knowledge, the authors' analysis and experimental setup is correct and sound. While the authors did not discuss the weaknesses of their algorithms except a brief argument in the conclusion, I am unable to identify any characteristic that requires any lengthy analysis.

2.3 Clarity

The article is clearly written with reasonable structure. The authors separated the theorem and lemma statements and the actual analysis, putting the outline of the analysis in the main article, while delaying the proofs to the appendix. This is a common practice, but it is arguable that structuring the proof, perhaps an abridged version under the statements, can better faciliate understanding.

Another critique is that many of the article's stated lemmas and theorems involves many variables and complicated mathematical expressions. While this complexity is justified by itself, there may be a lack of explanation around these statements to compensate for the complexity. This is especially relevant when the details of the proof are delayed to a later section. As a result, I find the lemmas and statements hard to comprehend without scrutiny. It would be better if the authors can append a sentence or two after statements with complex mathematical expressions, Theorem 2, 3, and 4 in the article, for example, to briefly explain the significance and meaning of these statements.

2.4 Significance

It is clear both from the authors' narration and the implications of their results that this article is of significance. Near neighbor search is a problem with many applications in various fields, and any improvement to methods solving the problem will imply more efficient and effective algorithms for all such fields. The authors clearly stated the improvements of FLINNG over other status quo algorithms and supported their claims with sound arguments, both theoretical and empirical. Their algorithms are also quite simplistic in nature, and although I am not experienced enough to immediately identify any potential extensions, no doubt future researchers can build upon this article for further improvements to near neighbor search or other significant problems.



**Time Spent Reviewing:**

3

---

> ### Author Response · Authors · 2021-08-10
> **Thank you for the thorough review**
>
> Thank you very much for your thoughtful review and helpful comments. We will add more introductory and explanatory comments in the proofs. It may be difficult to fit abridged versions of the proofs in the main text, but we'll try to add sentences that give more of a general intuition of the proof strategy. We also plan to include a table that compares the query conditions and time complexity of our work with other recent methods (both LSH-based and otherwise).

---

> > ### Comment · Reviewer_557Q · 2021-09-10
> > **response**
> >
> > I acknowledge reading the response and would be happy to see the promised edits being implemented. I maintain my score.

---

### Decision · Program_Chairs · 2021-09-27

**Decision:**

Accept (Spotlight)

**Comment:**

The submission was deemed significant, well-executed, and that it will have impact. While some reviewers had some reservations, the author rebuttals addressed the concerns.